# Joint Velocity-Growth Flow Matching for Single-Cell Dynamics Modeling

**Dongyi Wang[1],\*, Yuanwei Jiang[1],\*, Zhenyi Zhang[2],\*, Xiang Gu[1],†, Peijie Zhou[3],†, Jian Sun[1],†**

[1]School of Mathematics and Statistics, Xi'an Jiaotong University, Xi'an, China
[2]LMAM and School of Mathematical Sciences, Peking University, Beijing, China.
[3]Center for Machine Learning Research, Peking University, Beijing, China.
{dongyiwang,jyw1578857771}@stu.xjtu.edu.cn; {xianggu,jiansun}@xjtu.edu.cn;
zhenyizhang@stu.pku.edu.cn; pjzhou@pku.edu.cn

## Abstract

Learning the underlying dynamics of single cells from snapshot data has gained increasing attention in scientific and machine learning research. The destructive measurement technique and cell proliferation/death result in unpaired and unbalanced data between snapshots, making the learning of the underlying dynamics challenging. In this paper, we propose joint Velocity-Growth Flow Matching (VGFM), a novel paradigm that jointly learns state transition and mass growth of single-cell populations via flow matching. VGFM builds an ideal single-cell dynamics containing velocity of state and growth of mass, driven by a presented two-period dynamic understanding of the static semi-relaxed optimal transport, a mathematical tool that seeks the coupling between unpaired and unbalanced data. To enable practical usage, we approximate the ideal dynamics using neural networks, forming our joint velocity and growth matching framework. A distribution fitting loss is also employed in VGFM to further improve the fitting performance for snapshot data. Extensive experimental results on both synthetic and real datasets demonstrate that VGFM can capture the underlying biological dynamics accounting for mass and state variations over time, outperforming existing approaches for single-cell dynamics modeling. Our code is available at https://github.com/DongyiWang-66/VGFM.

## 1 Introduction

Inferring the latent dynamics of complex systems from sparse and noisy data is a fundamental challenge in science and engineering. In many domains, *e.g.*, stock markets [1], climate systems [2], and biological processes [3–11], continuous trajectories are rarely fully observed by sensors. Instead, cross-sectional snapshot data collected at discrete time points are commonly provided. This challenge is especially important in single-cell RNA sequencing [12–16], where destructive sampling yields unpaired population-level snapshots across time without having the tracked individual cell fates. Furthermore, due to the mass changes during cellular development or response processes, observed data often exhibits mass unbalancedness across time, violating mass conservation. Consequently, reconstructing the time-evolving, unnormalized density function from limited samples has become an important research problem and attracted increasing attention [14, 15, 17].

Deep learning-based models for dynamics inference have demonstrated great potential [3, 7, 18–29]. These models typically employ ordinary or stochastic differential equations (ODEs or SDEs) parameterized by neural networks to approximate the velocity field governing density evolution.

---

*These authors contributed equally. †Corresponding authors.

39th Conference on Neural Information Processing Systems (NeurIPS 2025).

One class of methods, based on simulation [3, 18, 19, 23, 24, 26, 30–32], generates synthetic trajectories by feeding initial data through neural networks and solving ODEs or SDEs numerically, and compares simulated results with observations for loss computation. However, these simulation-based methods rely heavily on numerical solvers during training, which significantly increases computational cost. In high-dimensional settings, the enlarged search space further exacerbates instability, hindering scalability and convergence. Another line of research focuses on simulation-free approaches [22, 29, 33–43], where the velocity field is trained efficiently by constructing conditional probability paths without trajectory simulation, yielding better efficiency and stability in training compared with simulation approaches.

Nevertheless, the simulation-free approaches mentioned above rely solely on the velocity field $v$, ignoring the unbalancedness of observed data, which can result in incorrect reconstruction of underlying dynamics or unsatisfying generation performance [26, 37, 42]. Distributional imbalance is a common phenomenon in single-cell dynamics, where cellular proliferation and death occur, thus necessitating the introduction of a growth term $g$ to allow for mass increasing or decreasing. To address this, [44, 45] proposed to jointly learn velocity $v$ and growth $g$ by minimizing the Wasserstein-Fisher-Rao (WFR) metric [46, 47]. However, it mathematically enforces $v = \nabla g$, which lacks clear mechanistic justification in biological systems. Other WFR-inspired approaches [23, 26] design separate neural networks for $v$ and $g$ and prioritize fitting the distributions. However, these methods require heavy simulation during training, limiting their scalability to high dimensions.

In this paper, we propose joint Velocity-Growth Flow Matching (VGFM), a novel approach for single-cell dynamics modeling. VGFM aims to learn the joint state transition[2] and mass growth[3] of single-cell evolution by flow matching, as illustrated in Fig. 1. VGFM is based on the static semi-relaxed optimal transport that allows mass variation in building coupling between snapshots, for which we present a dynamic understanding that enables mass growth and state transition accomplished in two time periods, respectively. Based on this dynamic understanding, we build a joint transition and growth dynamics, inheriting the optimal properties of semi-relaxed optimal transport.

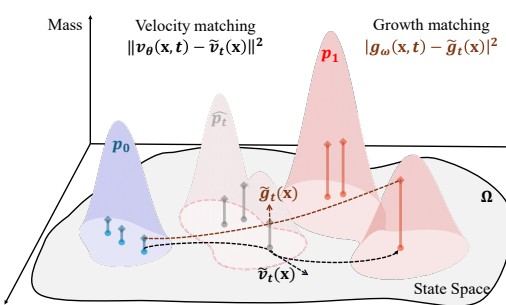

Figure 1: The goal of this paper is to learn a joint state transition (controlled by $v_\theta(\mathbf{x}, t)$) and mass growth (controlled by $g_\omega(\mathbf{x}, t)$) dynamics for single-cell evolution.

We then approximate the velocity and growth in the built dynamics using neural networks with finite samples, yielding a joint velocity and growth flow matching framework. To further improve fitting performance for the single-cell snapshots, we employ a distribution fitting loss based on the Wasserstein distance. Thanks to the joint velocity and growth flow matching, our approach enjoys more stable training and better scalability to high dimensions, compared with simulation-based approaches. Extensive experiments on both synthetic and real single-cell datasets are conducted to evaluate our proposed approach. Experimental results demonstrate accurate dynamics reconstruction and superior performance achieved by VGFM over recent baselines.

## 2 Background and Related Works

This section revisits optimal transport, and reviews the flow-matching-based methods for single-cell and unbalancedness-aware distribution learning methods, which are mostly related to this paper.

**Optimal transport.** Optimal transport, a tool for transforming one distribution into another at minimal cost [48], has gained remarkable prominence in recent years across fields such as image processing [49–54] and bioinformatics [18, 21, 22, 26, 41]. Given two distributions $p_0$ and $p_1$ on domain $\Omega$ satisfying $\int_\Omega p_0(\mathbf{x}) \, \mathrm{d}x = \int_\Omega p_1(\mathbf{x}) \, \mathrm{d}x$, the optimal transport [48] aims to find a joint distribution of $p_0$ and $p_1$, named coupling, such that the transport cost is minimized, formulated as

---

[2]State transition describes changes in gene expression in a cell.

[3]Mass growth means mass increasing or decreasing.

the following optimization problem:

$$\min_{\pi \in U(p_0, p_1)} \int_{\Omega^2} c(\mathbf{x}_0, \mathbf{x}_1) \, \mathrm{d}\pi(\mathbf{x}_0, \mathbf{x}_1), \text{ s.t. } U(p_0, p_1) = \left\{ \pi \geq 0 : \mathrm{P}^0_\# \pi = p_0, \, \mathrm{P}^1_\# \pi = p_1 \right\}, \quad (1)$$

where $c$ is the cost function. For a map $T$, we define $T_\# p_0(\mathbf{x}) = \int_{\mathbf{x}':T(\mathbf{x}')=\mathbf{x}} p_0(\mathbf{x}') \, \mathrm{d}\mathbf{x}'$. Let $\mathrm{P}^i(\mathbf{x}_0, \mathbf{x}_1) = \mathbf{x}_i$ for $i = 0, 1$, then we have $\mathrm{P}^i_\# \pi(\mathbf{x}_i) = \int_\Omega \pi(\mathbf{x}_0, \mathbf{x}_1) \, \mathrm{d}\mathbf{x}_i$ for $i = 0, 1$. When $c(\mathbf{x}_0, \mathbf{x}_1) = \|\mathbf{x}_0 - \mathbf{x}_1\|^2$, by Brenier's theorem [55], the optimal coupling can be expressed as $\pi^* = (\mathrm{Id}, T^*)_\# p_0$, where $T^*$ is called the Monge map.

**Flow matching for learning single cell dynamics.** Flow matching [33–35] is a simulation-free approach where the velocity field is trained efficiently by constructing conditional probability paths without trajectory simulation. Building on this foundation, further improvements have been achieved by incorporating optimal transport guidance [22, 36], considering unbalancedness [37], accounting for manifold structures [38], and approximating Schrödinger bridge via flow and score matching [41], all of which has been applied to model single-cell dynamics [22, 37, 38, 41]. Different from these methods, our method explicitly models the simultaneous matching of both the velocity field and the growth function, driven from our developed ideal joint state transition and mass growth dynamics, which does not require the construction of conditional probability paths like the above approaches.

**Unbalancedness-aware distribution learning methods.** Data imbalance is prevalent across a variety of domains, such as image synthesis and protein generation, motivating the development of flow matching models that account for unbalanced distributions [37, 42, 56]. However, these methods often do not model growth functions, which are essential in single-cell contexts. In the single-cell domain, several approaches have been proposed to model growth dynamics [5, 23, 44, 45, 57], yet they either fail to leverage the informative variations in cell abundance observable from snapshot data [44, 45], or rely heavily on computationally expensive simulations [5, 23, 57], limiting their scalability and efficiency. Differently, our method explicitly learns the growth function from observed snapshot data, yielding better performance for single cells than these methods as shown in experiments.

## 3 Method

Given the snapshot population of single cells, this paper aims to build the dynamic trajectory of single cells. Considering the unbalancedness between snapshot populations due to the undergo cell proliferation or death, we aim to build a model that can transform between unbalanced distributions by jointly learning a velocity field for controlling state transition and a growth function for controlling mass variation. Towards this goal, we propose joint Velocity-Growth Flow Matching (VGFM) based on semi-relaxed optimal transport to learn the dynamics of single cells. As illustrated in Fig. 2, we first present a decoupled understanding of state transition and mass growth for unbalanced dynamics based on semi-relaxed optimal transport (Figs. 2 (a), (b)), upon which we build a dynamic process between unbalanced distributions (Fig. 2 (c) ). Finally, we propose the velocity and growth flow matching using the built dynamic process to learn the velocity and growth with samples (Fig. 2 (d)). Next, we first describe the unbalanced dynamics of single-cell data, and then discuss the details of each component of our method.

### 3.1 Unbalanced Dynamics of Single Cell

Given two adjacent snapshots of populations/distributions denoted as $p_0$ and $p_1$, if the system is balanced, the dynamics $\bar{p}_t (t \in [0, 1])$ between $p_0$ and $p_1$ can be generated by ODE $\frac{\mathrm{d}\mathbf{x}_t}{\mathrm{d}t} = v_t(\mathbf{x}_t)$ such that $\mathbf{x}_0 \sim p_0$ and $\mathbf{x}_1 \sim p_1$. Corresponding to this ODE, $\bar{p}_t$ is governed by continuity equation $\partial_t \bar{p}_t = -\nabla \cdot (\bar{p}_t v_t)$. However, in biological systems, cells can proliferate and die. Such behaviors can be captured by a time-dependent weight $w_t(\mathbf{x}_t)$ associated with the state transition, whose evolution is controlled by a growth function $g$, simulating cell proliferation or death processes:

$$\begin{cases} \frac{\mathrm{d}\mathbf{x}_t}{\mathrm{d}t} = v_t(\mathbf{x}_t), \\ \frac{\mathrm{d}\log w_t(\mathbf{x}_t)}{\mathrm{d}t} = g_t(\mathbf{x}_t). \end{cases} \quad (2)$$

Under this formulation, the unbalanced distribution dynamics $p_t$ is generated jointly by the position $\mathbf{x}_t$ and the weight $w_t(\mathbf{x}_t)$. Specifically, $w_t(\mathbf{x}_t)$ models the variation of $p_t$ against $\bar{p}_t$, *i.e.*, $w_t(\mathbf{x}_t) =$

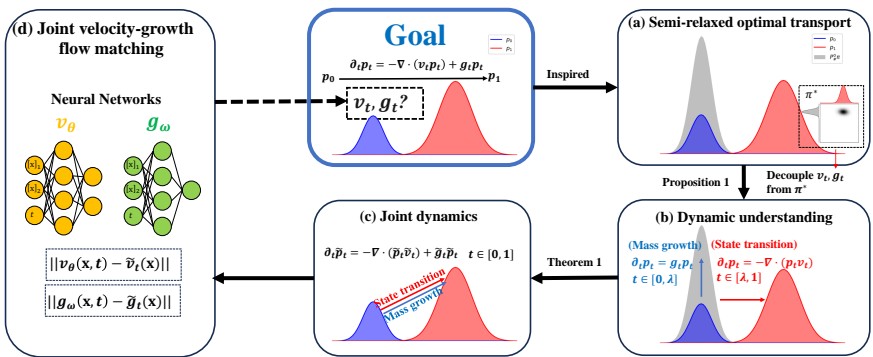

Figure 2: Illustration of our proposed VGFM, consisting of the velocity and growth flow matching deduced by the dynamic reformulation of the semi-relaxed optimal transport.

$p_t(\mathbf{x}_t)/\bar{p}_t(\mathbf{x}_t)$ for $\bar{p}_t(\mathbf{x}_t) \neq 0$. Corresponding to Eq. (2), $p_t$ is governed by

$$\partial_t p_t = -\nabla \cdot (p_t v_t) + g_t p_t. \tag{3}$$

## 3.2 Building Unlabalnced Dynamics Based on Semi-Relaxed Optimal Transport

Though Eqs. (2) or (3) provide a formulation of unbalanced dynamics between $p_0$ and $p_1$, the equations admit a wide range of admissible dynamics. We next seek the dynamics based on semi-relaxed optimal transport. Specifically, given the semi-relaxed optimal transport

$$\min_{\pi \geq 0} \mathcal{J}_{\mathrm{sot}}(\pi) \triangleq \int_{\Omega^2} c(\mathbf{x}_0, \mathbf{x}_1)\, \mathrm{d}\pi(\mathbf{x}_0, \mathbf{x}_1) + \mathrm{KL}(\mathrm{P}^0_{\#}\pi \| p_0) \quad \text{subject to} \quad \mathrm{P}^1_{\#}\pi = p_1, \tag{4}$$

between two unbalanced distributions $p_0$ and $p_1$, we will first present a dynamic understanding of Eq. (4) that decouples the state transition and mass growth into different periods. Based on this dynamic understanding, we will build a reasonable unbalanced dynamics between $p_0$ and $p_1$, which naturally drives our velocity and growth matching to learn the dynamics with samples (see Sect. 3.3).

**Dynamic understanding of semi-relaxed optimal transport.** As mentioned above, the solution of the semi-relaxed optimal transport allows both state transition and mass variation/growth. From a dynamic perspective, we understand the transport as a two-period dynamic process, in which we perform mass growth in the first period and state transition in the second period. Note that such an understanding is not practical in applications, but it provides a decoupled modeling of state transition and mass growth, which promotes a reasonable and friendly-to-matching unbalanced dynamics modeling, driving our flow matching algorithm for learning velocity and growth, as we will make clear later. More concretely, given $\lambda \in (0, 1)$, we perform mass growth controlled by $g_t$ for $t \in [0, \lambda]$ through $\partial_t p_t = g_t p_t$ with initial and ending distributions of $p_0$ and $p_\lambda$ respectively. For $t \in (\lambda, 1]$, the state transition controlled by $v_t$ through $\partial_t p_t = -\nabla \cdot (p_t v_t)$ is performed with initial and ending distributions of $p_\lambda$ and $p_1$, respectively. We then define the following two-period transport model:

$$\min_{(v_t, g_t) \in \mathcal{C}_\lambda(p_0, p_1)} \mathcal{J}^\lambda_{\mathrm{tpt}}(v_t, g_t) \triangleq (1 - \lambda) \int_\Omega \int_\lambda^1 p_t(\mathbf{x}) \|v_t(\mathbf{x})\|^2 \mathrm{d}t \mathrm{d}\mathbf{x} + \mathcal{H}(v_t, g_t, p_t), \tag{5}$$

where $\mathcal{H}(v_t, g_t, p_t) = \int_\Omega p_0(\mathbf{x})(e^{\int_0^\lambda g_t(\mathbf{x})\mathrm{d}t}(\int_0^\lambda g_t(\mathbf{x})\mathrm{d}t - 1) + 1)\mathrm{d}\mathbf{x}$, $\mathcal{C}_\lambda(p_0, p_1) = \{(v_t, g_t) : \partial_t p_t = g_t p_t, t \in [0, \lambda]; \partial_t p_t = -\nabla \cdot (p_t v_t), t \in (\lambda, 1]\}$, and $p_0$ and $p_1$ are the given distributions. The following proposition shows the relation between the semi-relaxed optimal transport model in Eq. (4) and the two-period transport model in Eq. (5).

**Proposition 1.** *Assume $c(\mathbf{x}_0, \mathbf{x}_1) = \|\mathbf{x}_0 - \mathbf{x}_1\|^2$ and if we enforce $\mathrm{P}^0_{\#}\pi$ and $p_0$ to share the same support for admissible solution $\pi$ to problem (4), then we have $\min_\pi \mathcal{J}_{\mathrm{sot}}(\pi) = \min_{v_t, g_t} \mathcal{J}^\lambda_{\mathrm{tpt}}(v_t, g_t), \forall \lambda \in (0, 1)$. Moreover, for any $\lambda \in (0, 1)$, given the optimal transport plan $\pi^*$ to problem (4), let $p^*_\lambda \triangleq \mathrm{P}^0_{\#}\pi^*$, then $\pi^*$ can be expressed as $\pi^* = (\mathrm{Id}, T^*)_{\#}p^*_\lambda$ where $T^*$ is the Monge map between $p^*_\lambda$ and $p_1$. Meanwhile, there exist a $g^*_t$ such that $p^*_\lambda = p_0(\mathbf{x})e^{\int_0^\lambda g^*_t(\mathbf{x})\mathrm{d}t}$, and a $v^*_t$ given by $v^*_t\left(\mathbf{x} + \frac{t-\lambda}{1-\lambda}(T^*(\mathbf{x}) - \mathbf{x})\right) = \frac{T^*(\mathbf{x}) - \mathbf{x}}{1-\lambda}$, satisfying $(v^*_t, g^*_t) \in \arg\min_{v_t, g_t} \mathcal{J}^\lambda_{\mathrm{tpt}}(v_t, g_t)$.*

The proof is provided in the Appendix A.2. Proposition 1 indicates that problems (4) and (5) share the same optimal objective function value. Meanwhile, the optimal objective function value of the two-period transport problem (5) is independent of $\lambda$. Moreover, $v_t^*, g_t^*$ can be constructed using $\pi^*$.

**Building unbalanced dynamics.** Although we have identified a scheme in which $p_0$ evolves into $p_1$ through a two-period process governed separately by $v_t$ and $g_t$ respectively, this decoupling does not align with the behavior observed in biological systems, where transport and growth typically occur simultaneously. To address this, we next build an unbalanced dynamics based on the dynamic understanding discussed above that allows velocity and growth to jointly drive the state transition and mass growth while ensuring consistency with the target distribution. Specifically, given $v_t$ ($t \in (\lambda, 1]$), $g_t$ ($t \in [0, \lambda]$) of the two-period process, we define the joint velocity $\tilde{v}_t$ and growth $\tilde{g}_t$ by

$$
\begin{aligned}
\tilde{v}_t(\mathbf{x}) &= (1 - \lambda) \cdot v_{(1-\lambda)t+\lambda}(\mathbf{x}), \\
\tilde{g}_t(\mathbf{x}) &= \lambda \cdot g_{\lambda t} \left( \psi_{\tilde{v},t}^{-1}(\mathbf{x}) \right),
\end{aligned}
\tag{6}
$$

where $t \in [0, 1]$, and $\psi_{\tilde{v},t}$ is the flow generated by $\tilde{v}$, i.e., $\psi_{\tilde{v},t}(\mathbf{x}_0) = \mathbf{x}_0 + \int_0^t \tilde{v}_s(\mathbf{x}) \mathrm{d}s$.

**Theorem 1.** *Given the initial distribution $p_0$, denote the ending distribution of the two-period dynamics*

$$
\partial_t p_t = g_t p_t, t \in [0, \lambda]; \quad \partial_t p_t = -\nabla \cdot (p_t v_t), t \in (\lambda, 1],
\tag{7}
$$

*as $p_1$, and denote the ending distribution of the joint dynamics starting from $p_0$*

$$
\partial_t \tilde{p}_t = -\nabla \cdot (\tilde{p}_t \tilde{v}_t) + \tilde{g}_t \tilde{p}_t, \quad t \in [0, 1], \quad \tilde{p}_0 = p_0,
\tag{8}
$$

*as $\tilde{p}_1$, then it holds that $\tilde{p}_1 = p_1$.*

The proof is detailed in Appendix A.3. Theorem 1 indicates that given the same initial distribution at $t = 0$, the two-period and the joint dynamics yield the same distribution at $t = 1$. Since $\tilde{v}_t, \tilde{g}_t$ are defined on $[0, 1]$, the joint dynamics is more practical in applications, e.g., biological systems. Meanwhile, if $v_t, g_t$ is defined as in Proposition 1, $\tilde{v}_t, \tilde{g}_t$ will inherit the optimality properties of the semi-relaxed optimal transport, benefitting our flow matching algorithm, as shown in Sect. 3.3.

### 3.3 Velocity and Growth Flow Matching

Since the optimal growth function $g_t^*$ in Proposition 1 is required to evolve $p_0$ to $\mathrm{P}_\#^0 \pi^*$, satisfying the marginal constraint $p_0 \exp \left( \int_0^\lambda g_s^* \, ds \right) = \mathrm{P}_\#^0 \pi^*$, possible choice for $g_t^*$ is not unique. For simplicity and ease of implementation, we choose a time-independent form: $g_t^*(\mathbf{x}) = \frac{\log \mathrm{P}_\#^0 \pi^*(\mathbf{x}) - \log p_0(\mathbf{x})}{\lambda}$, which minimizes the $L^2$-norm energy functional of $g$ and can be explained by the Malthusian growth model [58], i.e., the exponential growth model (see Appendix A.4). By plugging the expression of $v_t^*$ and consists with from Proposition 1 and $g_t^*$ into Eq. (6), we obtain

$$
\tilde{v}_t(\psi_{\tilde{v},t}(\mathbf{x}_0)) = T^*(\mathbf{x}_0) - \mathbf{x}_0, \quad \tilde{g}_t(\psi_{\tilde{v},t}(\mathbf{x}_0)) = \log \mathrm{P}_\#^0 \pi^*(\mathbf{x}_0) - \log p_0(\mathbf{x}_0), \text{ where } \mathbf{x}_0 \sim p_0. \tag{9}
$$

In practice, only finite samples of $p_0$ and $p_1$ can be accessed. We then aim to learn neural networks $v_\theta(\mathbf{x}, t), g_\omega(\mathbf{x}, t)$ to approximate $\tilde{v}_t(\mathbf{x}), \tilde{g}_t(\mathbf{x})$ by velocity and growth flow matching as

$$
\min_{\theta, \omega} \mathbb{E}_{\mathbf{x}_0} \mathbb{E}_t \left[ \|v_\theta(\psi_{\tilde{v},t}(\mathbf{x}_0), t) - \tilde{v}_t(\psi_{\tilde{v},t}(\mathbf{x}_0))\|^2 + |g_\omega(\psi_{\tilde{v},t}(\mathbf{x}_0), t) - \tilde{g}_t(\psi_{\tilde{v},t}(\mathbf{x}_0))|^2 \right], \tag{10}
$$

that can generalize to new samples. We first estimate $\pi^*$ and $T^*$ using samples. Given samples $\{\mathbf{x}_0^i\}_{i=1}^n$ of $p_0$ and $\{\mathbf{x}_0^j\}_{j=1}^m$ of $p_1$, We seek the optimal transport plan $\pi^{0 \to 1} \approx \pi^*(\mathbf{x}_0, \mathbf{x}_1)$ by solving the entropy-regularized semi-relaxed transport problem using Sinkhorn-based algorithm [59, 60] as

$$
\pi^{0 \to 1} = \arg\min_{\pi \geq 0} \sum_{i,j} c_{ij} \pi_{ij} + \epsilon H(\pi) + \tau \mathrm{KL}(\pi \mathbf{1}_m \| \mathbf{1}_n), \quad \text{subject to} \quad \pi^\top \mathbf{1}_n = \mathbf{1}_m, \tag{11}
$$

where $c_{ij} = \|\mathbf{x}_0^i - \mathbf{x}_1^j\|^2$, $\mathbf{1}_m$ is the all-one vector, $H(\pi)$ denotes the negative entropy of $\pi$, and $\epsilon$ and $\tau$ are hyperparameters. With $\pi^{0 \to 1}$, $T^*$ can be estimated using barycentric mapping $T^*(\mathbf{x}_0^i) \approx \frac{1}{N^i} \sum_j \pi_{ij}^{0 \to 1} \mathbf{x}_1^j$ where $N^i = \sum_j \pi_{ij}^{0 \to 1}$. In implementation, for each $\mathbf{x}_0^i$, we sample $j$

from $(1, 2, \cdots , m)$ with probability $\frac{1}{N^i}\pi_{ij}^{0 \to 1}$, and approximate $T^*(\mathbf{x}_0^i) \approx \mathbf{x}_1^j$. This approximation is accurate as $\epsilon \to 0$. Therefore, $\psi_{\tilde{v},t}(\mathbf{x}_0^i) \approx \mathbf{x}_0^i + t(\mathbf{x}_1^j - \mathbf{x}_0^i) \triangleq \mathbf{x}_t$ for $t \in [0, 1]$, and Eq. (10) becomes

$$\mathcal{L}_{\text{VGFM}}(\theta, \omega) = \sum_{i=1}^{n} \sum_{j=1}^{m} \pi_{ij}^{0 \to 1} \mathbb{E}_t \left[ \left\| v_\theta(\mathbf{x}_t, t) - (\mathbf{x}_1^j - \mathbf{x}_0^i) \right\|^2 + \left| g_\omega(\mathbf{x}_t, t) - \log([\pi^{0 \to 1}\mathbf{1}_m]_i) \right|^2 \right],$$
(12)

where $[\cdot]_i$ is the $i$-th element. The last term in Eq. (12) is because $p_0(\mathbf{x}_0^i) = 1$, for $i = 1, \cdots , n$. In experiments, $\mathcal{L}_{\text{VGFM}}$ can be implemented using mini-batch samples. Specifically, we sample $(i, j)$ from $\pi^{0 \to 1}$ and $t$ from $\mathcal{U}(0, 1)$, then calculate the loss in square brackets. To improve robustness, we add a Gaussian noise to $\mathbf{x}_t$ in experiments.

### 3.4 Training Process

Our ultimate goal is to generate data close to the real data distribution. The current learned velocity field of flow matching is the expectation of the conditional velocity field, *i.e.*, $v_\theta(x, t) = \mathbb{E}_{z|(x,t)} v_\theta(x, t|z)$. In practice, since we use limited samples to learn the velocity field $v$, each numerical integration step introduces an approximation error, which can be accumulated during the numerical ODE solving process, resulting in a deviation from the true trajectory [61]. To address this bias, we employ distribution fitting loss besides the flow matching loss in Eq. (12), to improve performance further. Next, we introduce the distribution fitting loss and our training algorithm.

To achieve better performance, we incorporate the Wasserstein distance between generated samples and observed samples as part of the loss function. Specifically, let $\mathcal{X}_t = \{\mathbf{x}_t^i\}_{i=1}^{N_t}$ for $t = 0, 1, \cdots , T-1$ denote the observed samples at different time points, and denote $p(\mathcal{X}_t) = \sum_{i=1}^{N_t} \delta_{\mathbf{x}_t^i}$. We define $\phi_{v_\theta} : \mathbb{R}^d \times \mathcal{T} \to \mathbb{R}^{d|\mathcal{T}|}$ as the trajectory mapping function parameterized by the neural network $v_\theta$, which takes a given starting point as the initial condition and outputs particle coordinates at time indices $\mathcal{T}$ according to ODE dynamics, where $\mathcal{T}$ denotes a set of time steps. Given an initial set $\mathcal{X}_0$, the model $\phi_{v_\theta}$ predicts particle positions at future time points in $\mathcal{T}$. Specifically, the predicted samples are computed by applying the neural ODE to $\mathcal{X}_0$ over the time indices $\{1, \cdots , T-1\}$, *i.e.*, $\hat{\mathcal{X}}_1, \cdots , \hat{\mathcal{X}}_{T-1} = \phi_{v_\theta}(\mathcal{X}_0, \{1, \cdots , T-1\})$. Similarly, we define $\phi_{g_\omega}$ as the particle weight mapping function parameterized by the neural network $g_\omega$, which takes the initial weight of particle $i$ as input and outputs the corresponding weight values at $\mathcal{T}$ under ODE dynamics. For simplicity of notation, we use $\hat{w}(\hat{\mathbf{x}})$ for $\hat{\mathbf{x}} \in \hat{\mathcal{X}}_t$ to denote the weight generated by $\phi_{g_\omega}$ corresponding to the particles in set $\hat{\mathcal{X}}$ which is generated by $\phi_{v_\theta}$. The distribution fitting loss is defined as the Wasserstein distance as

$$\mathcal{L}_{\text{OT}}(\theta, \omega) = \sum_{t=1}^{T-1} \mathcal{W}_1 \left( \frac{1}{N_t} p(\mathcal{X}_t), \frac{1}{\sum_{\hat{\mathbf{x}} \in \hat{\mathcal{X}}_t} \hat{w}(\hat{\mathbf{x}})} p_{\hat{w}}(\hat{\mathcal{X}}_t) \right),$$
(13)

where $p_{\hat{w}}(\hat{\mathcal{X}}_t) = \sum_{\hat{\mathbf{x}} \in \hat{\mathcal{X}}_t} \hat{w}(\hat{\mathbf{x}}) \delta_{\hat{\mathbf{x}}}$, $\mathcal{W}_1$ is the 1-Wasserstein distance with Euclidean norm.

**Total loss and algorithm.** Note that the matching loss can be easily generalized to multi-time snapshots by employing Eq. (12) among two consecutive snapshots $p_t$ and $p_{t+1}$. Combined with distribution fitting loss in Eq. (13), we obtain the final loss

$$\mathcal{L}(\theta, \omega) = \mathcal{L}_{\text{VGFM}}(\theta, \omega) + \mathcal{L}_{\text{OT}}(\theta, \omega)$$
(14)

During training, we employ a parameter scheduling scheme in which we initially use $\mathcal{L}_{\text{VGFM}}$ as a warm-up stage, before enabling both loss terms for joint training. This design provides a notable advantage: after the warm-up stage, $v_\theta$ and $g_\omega$ are well-initialized through conditional matching, which facilitates convergence when switching to $\mathcal{W}_1$-based training. The detailed algorithm is presented in Algorithm 1. For further details on the warm-up procedure and parameter scheduling, please refer to Appendix B.1.

## 4 Experiments

We evaluate VGFM on synthetic and real datasets, and compare it with state-of-the-art approaches, including methods that account for distributional unbalancedness [23, 26, 57] as well as those that assume mass conservation [22, 38, 41]. Our code will be released online.

**Algorithm 1:** Training algorithm of joint velocity-growth flow matching

---

**Input:** Observed data $\mathcal{X}_0, \ldots, \mathcal{X}_{T-1}$; warm-up epochs $M_1$; training epochs $M_2$
**Output:** Trained velocity field $v_\theta$ and growth function $g_\omega$
Compute transport plans $\pi^{t \to t+1}$ for $t = 0, \ldots, T-2$;
**for** $i = 1$ **to** $M_2$ **do**
    **for** $t_0 = 0$ **to** $T - 2$ **do**
        Sample a batch $(i, j) \sim \pi^{t_0 \to t_0+1}$;
        Sample $t \sim \mathcal{U}(0,1) + t_0$, and $\mathbf{x}_t \sim \mathcal{N}(t\mathbf{x}_1^j + (1-t)\mathbf{x}_0^i, \sigma^2 I)$;
        $\mathcal{L} \leftarrow \mathcal{L} + \left\| v_\theta(\mathbf{x}_t, t) - (\mathbf{x}_1^j - \mathbf{x}_0^i) \right\|^2 + \left| g_\omega(\mathbf{x}_t, t) - \log([\pi^{0 \to 1}\mathbf{1}_m]_i) \right|^2$;
        **if** $i > M_1$ **then**
            $\hat{\mathcal{X}}_{t_0+1} \leftarrow \phi_{v_\theta}(\hat{\mathcal{X}}_{t_0}, t_0 + 1), \quad \hat{w}_{t_0+1}(\hat{\mathcal{X}}_{t_0+1}) \leftarrow \phi_{g_\omega}(\hat{\mathcal{X}}_{t_0}, t_0 + 1)$;
            $\mathcal{L} \leftarrow \mathcal{L} + \mathcal{W}_1\left( \frac{1}{N_{t_0+1}} p(\mathcal{X}_{t_0+1}), \frac{1}{\sum_{\hat{\mathbf{x}} \in \hat{\mathcal{X}}_t} \hat{w}(\hat{\mathbf{x}})} p_{\hat{w}}(\hat{\mathcal{X}}_{t_0+1}) \right)$;
    Update $\theta, \omega$ using $\mathcal{L}$

---

**Synthetic datasets.** Inspired by [23, 26], we adopt the Simulation Gene dataset that applies a three-gene regulatory network to produce a quiescent region and an area exhibiting both transition and growth that can be observed. We also use Dyngen [62] to simulate a scRNA-seq dataset from a dynamic cellular process, which exhibits pronounced branching unbalancedness, with significantly different cell abundances across divergent lineages. Following the experiment setup of [31], we use PHATE [63] to reduce its dimensions to 5. Additionally, inspired by [26] that employs a high-dimensional Gaussian mixture model [64] to evaluate the scalability of models, we adopt a more challenging setting: 1000-dimensional Gaussian mixtures.

**Real-world dataset.** We conduct experiments on three real-world datasets, Embroid Body (EB) [65], CITE-seq (CITE) [66] and Pancreas [67], preprocessed following the procedures in [18, 22, 37]. EB and CITE dataset are evaluated under both 5- and 50-dimensional PCA projections, while Pancreas dataset is evaluated under 2000 highly variable gene space to assess VGFM's scalability on real-world data. Specifically, the EB (5D), CITE (5D), and CITE (50D) configurations are assessed using a hold-out strategy, same as [38], in which an intermediate time point is excluded during training. The model is then used to predict the distribution at the hold-out time, and we compute the $\mathcal{W}_1$ distance between the predicted and true distributions at that time point. The EB (50D) setting is further used for comparison against [26] and ablation study, evaluated using both $\mathcal{W}_1$ and the Relative Mass Error (RME) (see below). In the Pancreas dataset, in addition to $\mathcal{W}_1$ and RME, we also present the mean–variance trends between generated and real gene expressions and analyze the interpretability of the learned growth function $g$.

**Evaluation metrics.** We follow [26] to use $\mathcal{W}_1$ to measure the distance between the predicted cell distribution and the true cell distribution. Additionally, to assess the accuracy of the growth function $g_\omega$, we normalize the total mass at time 0 to 1, and denote the relative mass at time $t$ *w.r.t* time 0 as $m_t$, *i.e.*, $m_t = \frac{N_t}{N_0}$. The predicted relative mass at time $t$ based on the evolution induced by $g_\omega$ is denoted as $\hat{m}_t = \sum_{\hat{\mathbf{x}} \in \hat{\mathcal{X}}_t} \hat{w}(\hat{\mathbf{x}})/N_0$, where $\sum_{\hat{\mathbf{x}} \in \hat{\mathcal{X}}_t} \hat{w}(\hat{\mathbf{x}})$ is the predicted total mass at time $t$. We define the relative mass error, denoted as RME, *i.e.*, $\text{RME} = \frac{|m_t - \hat{m}_t|}{m_t}$.

**Implementation details.** We employ a 3-layer (5-layer for dimensions greater than 50) MLP with 256 hidden units and LeakyReLU activation to parameterize both the velocity field $v_\theta$ and the growth function $g_\omega$. Optimization is performed using the Adam optimizer with a learning rate of $10^{-3}$ at warm-up stage and $10^{-4}$ after applying distribution fitting loss. A warm-up stage of 500 iterations for synthetic datasets and 5000 iterations for real-world dataset is applied (only matching loss in Eq. (12) ), after which the distribution fitting loss $\mathcal{L}_{\text{OT}}$ defined in Eq. (13) is applied for an additional 30 training epochs. We will discuss the strategy to set $\epsilon$ and $\tau$ in the Appendix B.1.

## 4.1 Results and Analysis

**Ability to reconstruct cellular dynamics.** As shown in Tab. 1, our model achieves lowest mean $\mathcal{W}_1$ and RME on synthetic datasets, outperforming both unbalancedness-aware methods [23, 26, 57] and

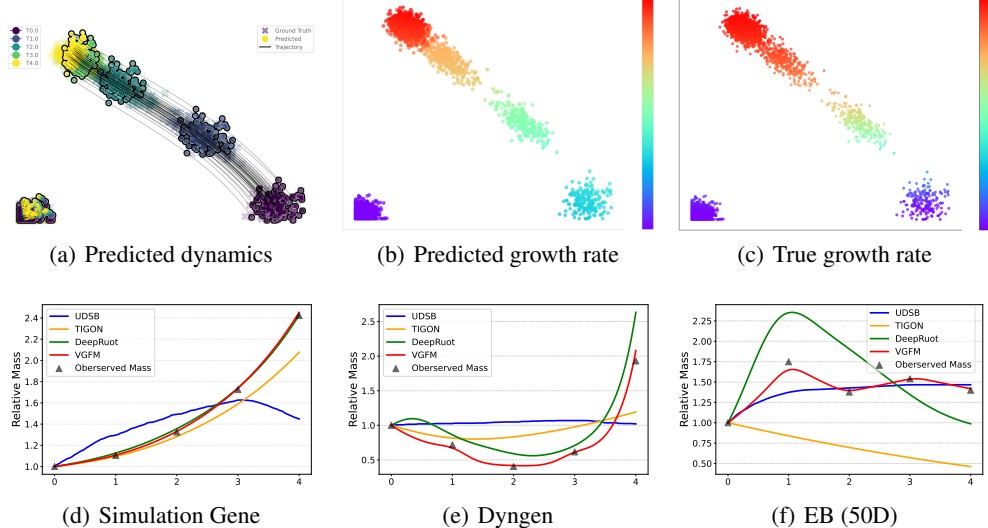

(a) Predicted dynamics     (b) Predicted growth rate     (c) True growth rate

(d) Simulation Gene     (e) Dyngen     (f) EB (50D)

Figure 3: (a) Predicted dynamics with trajectories of VGFM on Simulation Gene. (b) Predicted and (c) true growth rates on Simulation Gene. Note that for Simulation Gene, the true growth rates could be accessed. (d), (e) and (f) respectively compare predicted relative mass by UDSB, TIGON, DeepRUOT, and VFGM, and observed relative mass from data, on three datasets. Note that in (f), on 50D dataset, the mass predicted by TIGON deviates significantly from the observed trend, because of its difficulty in handling high-dimensional data as discussed in their paper [23].

Table 1: Mean $\mathcal{W}_1$ and RME over all time on synthetic datasets. *OT-CFM and OT-MFM do not model the growth function, thus, RME is not computed. "N/C" means "not converge".

| Method | Simulation Gene (2D) | | Dyngen (5D) | | Gaussian (1000D) | |
|---|---|---|---|---|---|---|
| | $\mathcal{W}_1$ | RME | $\mathcal{W}_1$ | RME | $\mathcal{W}_1$ | RME |
| OT-CFM* [22] | 0.302 | — | 3.926 | — | 10.126 | — |
| OT-MFM* [38] | 0.311 | — | 3.976 | — | 11.008 | — |
| UDSB [57] | 0.665 | 0.192 | 1.914 | 0.658 | N/C | N/C |
| TIGON [23] | 0.099 | 0.065 | 1.029 | 0.542 | N/C | N/C |
| DeepRUOT [26] | 0.068 | 0.016 | 0.474 | 0.199 | N/C | N/C |
| **VGFM** | **0.046** | **0.006** | **0.420** | **0.053** | **3.010** | **0.037** |

unbalancedness-ignoring methods [22, 38], which demonstrate the ability of VGFM to model cellular dynamics for data and mass fitting. We also visualize the learned trajectory and growth function on the Simulation Gene dataset in Figs. 3 (a-c). Figs. 3 (a-c) show that VGFM not only reconstructs accurate trajectories but also produces faithful predictions of spatially varying growth rates.

**Scalability and mass-matching performance compared with existing unbalanced models.** While existing SOTA approaches of UDSB [57], TIGON [23] and DeepRUOT [26] incorporate unbalanced modeling, these models either introduce biologically implausible transitions [57] (as discussed in [26]) or face difficulties in scaling to high-dimensions [23, 26, 57]. As a result, these methods do not perform well or converge on the 1000-dimensional Gaussian dataset, as in Tab. 1. In contrast, VGFM integrates flow matching based on unbalanced transport, allowing it to scale more effectively to high-dimensional datasets. Our method more accurately recovers the ground-truth dynamics defined in [26] and achieves superior mass-matching accuracy, as shown in Tab. 1 and Figs. 3 (d-f).

**Hold-one-out results on real-world datasets.** Table 2 compares hold-one-out results of different methods on EB (5D) and CITE (5D and 50D) datasets. It can be observed that VGFM outperforms most existing approaches in terms of $\mathcal{W}_1$ distance. We attribute this to the integration of complementary strengths from both flow matching and simulation-based (Neural ODE) frameworks. On the one hand, flow matching provides a robust initialization for learning the velocity field and growth rate,

which helps maintain training stability after incorporating the distribution fitting loss $\mathcal{L}_{\text{OT}}$, leading to better performance than purely simulation-based models, as also shown in Fig. 4. On the other hand, since the flow matching objective serves only as an upper bound on the true Wasserstein distance [68], the introduction of $\mathcal{L}_{\text{OT}}$ allows direct optimization of the distribution fitting loss, further enhancing model performance. Moreover, by explicitly modeling cell growth through a time-varying weight function, our framework generalizes beyond the conventional setting where Wasserstein distance is computed between unweighted discrete measures, thereby offering greater flexibility in evaluation and enabling more possible transport solutions.

**Pancreas dataset under 2000-dimensional gene space.** To further explore the scalability of VGFM, we applied our model and compared methods explicitly modeling $g_t(x)$ [23, 26] to the Pancreas dataset with 2000 highly variable genes. We first observed that VGFM is the only method showing a steadily decreasing training loss, both for $\mathcal{L}_{\text{VGFM}}$ and $\mathcal{L}_{\text{OT}}$, as shown in Fig.A-11. Next, following [69], we calculated the means and variances of the real and generated gene at day 15.5 and plotted the corresponding mean-variance trend and histograms. The results show that the generated samples closely follow the real data, as shown in Fig. A-15. Finally, we analyzed the learned $g_w(x, t)$ and found that our model successfully reconstructs the unbalanced pattern and identifies key genes without being given any prior knowledge of the cell types. The results are reported in Fig. A-16, A-17 and A-18. For more details, please refer to Appendix B.8.

**Ablation study on loss terms.** To assess the contribution of each loss component in our framework, we perform an ablation study on the EB dataset with unwhitened 50-dimensional PCA features. Table 3 reports the performance across four time points. VGFM in general achieves the lowest $\mathcal{W}_1$ and RME values (excluding time 1) compared with VGFM (w/o $\mathcal{L}_{\text{OT}}$) and VGFM (w/o $\mathcal{L}_{\text{VGFM}}$), demonstrating the effectiveness of both our joint velocity-growth matching loss and distribution fitting loss. Notably, by removing $\mathcal{L}_{\text{VGFM}}$, both $\mathcal{W}_1$ and RME increase by a significant margin, implying the importance of our velocity and growth matching to our framework. Consistent with our motivation in Sect. 3.4, adding in distribution fitting loss $\mathcal{L}_{\text{OT}}$ results in lower $\mathcal{W}_1$ and RME, improved fitting ability to observed distributions.

Table 2: Mean hold-one-out results on EB and CITE datasets over hold-out times.

| **Method** | EB | CITE | |
| --- | --- | --- | --- |
| | 5D | 5D | 50D |
| OT-CFM [22] | 0.790 | 0.882 | 38.756 |
| SF$^2$M [41] | 0.793 | 0.920 | 38.524 |
| UDSB [57] | 1.206 | 2.023 | 44.168 |
| OT-MFM [38] | 0.713 | **0.724** | **36.394** |
| DeepRUOT [26] | 0.774 | 0.845 | 38.681 |
| **VGFM** | **0.676** | 0.745 | 37.386 |

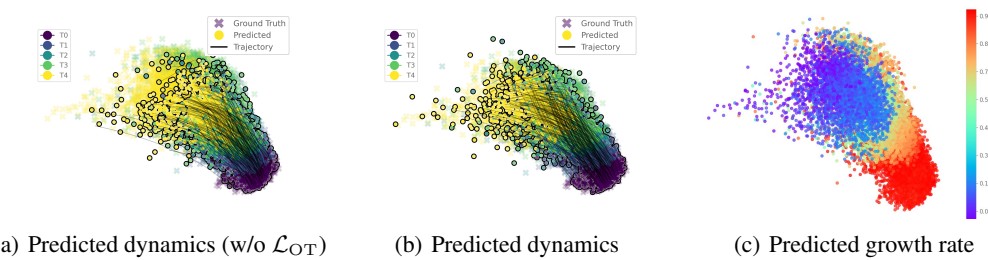

(a) Predicted dynamics (w/o $\mathcal{L}_{\text{OT}}$)    (b) Predicted dynamics    (c) Predicted growth rate

Figure 4: Visualization of predicted dynamics by (a) VFGM (w/o $\mathcal{L}_{\text{OT}}$) and (b) VGFM, and growth rate by (c) VFGM, on EB (5D) dataset, where the hold-out time is the first intermediate timepoint.

**Comparison on computational efficiency.** We compare the computational cost of our method and the SOTA method DeepRUOT [26] in Tab. 3. For fair comparison, all methods in Tab. 3 adopt the same networks 5-layer MLP with 256 hidden units, and are trained on the same GPU until convergence. As shown in Table 3, our full model requires slightly more training time than the variant using only $\mathcal{L}_{\text{VGFM}}$, due to the training with additional distribution fitting loss. However, it still achieves faster convergence compared with simulation-based methods of DeepRUOT [26] and the ablation variant without $\mathcal{L}_{\text{VGFM}}$. This demonstrates the critical role of the loss $\mathcal{L}_{\text{VGFM}}$ for facilitating training convergence.

To make $\mathcal{L}_{\mathrm{OT}}$ a fully differentiable component in our training pipeline, we also employed the Sinkhorn divergence [70] to replace $\mathcal{W}_1$ as the distribution fitting loss $\mathcal{L}_{\mathrm{OT}}$, forming a variant of VGFM denoted as VGFM(*) in Tab. 3. It can be observed that using the Sinkhorn divergence leads to a further improvement in the accuracy of dynamics reconstruction, while reducing the computational cost for distribution fitting. For more details, please refer to Appendix B.6.

Table 3: Ablation study on EB (50D) dataset. For comparison, we also report the results of the SOTA approach DeepRUOT [26] in this table.

| Method | $t_1$ | | $t_2$ | | $t_3$ | | $t_4$ | | Training time |
|---|---|---|---|---|---|---|---|---|---|
| | $\mathcal{W}_1$ | RME | $\mathcal{W}_1$ | RME | $\mathcal{W}_1$ | RME | $\mathcal{W}_1$ | RME | |
| DeepRUOT [26] | 8.169 | 0.416 | 9.041 | 0.415 | 9.348 | 0.119 | 9.808 | 0.296 | 90 (mins) |
| VGFM (w/o $\mathcal{L}_{\mathrm{OT}}$) | 8.915 | 0.020 | 10.590 | 0.098 | 10.915 | 0.067 | 11.635 | 0.088 | 6 (mins) |
| VGFM (w/o $\mathcal{L}_{\mathrm{VGFM}}$) | 8.644 | 0.650 | 10.167 | 0.710 | 11.052 | 0.823 | 11.530 | 0.862 | 62 (mins) |
| VGFM | 7.951 | 0.089 | **8.747** | 0.042 | 9.244 | **0.019** | 9.620 | **0.044** | 13 (mins) |
| VGFM (*) | **7.902** | **0.018** | 8.767 | **0.013** | **9.063** | 0.083 | **9.507** | 0.096 | 9 (mins) |

## 5 Conclusion, Limitation, and Future Work

The paper proposes the joint Velocity-Growth Flow Matching (VGFM) method that jointly learns state transition and mass growth of single-cell populations via flow matching. VGFM designs an ideal dynamics containing a velocity field and a growth function, driven by our presented two-period dynamic understanding of the static semi-relaxed optimal transport model. Approximating the ideal dynamics using neural networks yields the velocity-growth joint flow matching framework.

Although our approach achieves better scalability and training efficiency compared to simulation-based methods, it is not entirely simulation-free due to the incorporation of the distribution fitting loss $\mathcal{L}_{\mathrm{OT}}$. Moreover, since the learned growth rate relies on the observed cell counts at each time point, it may inadvertently capture effects that are not solely attributable to biological growth. Future work could address them by integrating biological priors into the learning process of the growth function $g$, and by exploring fully simulation-free alternatives that can achieve comparable performance.

## Acknowledgment

This work was supported by the National Key R&D Program 2021YFA1003002, NSFC (12125104, 623B2084, 12501709, 12426313), China National Postdoctoral Program for Innovative Talents (BX20240276), China Fundamental Research Funds for the Central Universities (xzy022025047) and China Postdoctoral Science Foundation (2025M773058).

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

## Outline of Appendix

The appendix is organized into three main parts.

Appendix A presents the theoretical foundations. We begin by reviewing the Brenier–Benamou theorem [71], followed by detailed proofs of Proposition 1 and Theorem 1 that are stated in the main text, as well as the justification for our choice of the reparameterized growth function $\tilde{g}$ in Eq. (9).

Appendix B focuses on experiments. We first describe our general implementation setup, including the strategy for selecting the hyperparameters in Eq. (11). Next, we provide detailed settings, visualizations and analysis for each experiment.

Appendix C discusses the broader impacts of our work.

## A   Proofs

### A.1   Background: Brenier-Benamou Formulation

We first recall the dynamic formulation of balanced optimal transport, which is known as Brenier-Benamou formula [71]. In this paper, we assume $p_0$ and $p_1$ are absolutely continuous *w.r.t.* Lebesgue measure, which is omitted for convenience of description.

**Theorem A-2** (Brenier-Benamou formula). *Given two probability measures $p_0, p_1 \in \mathcal{P}_2(\Omega)$, it holds that*

$$\mathcal{W}_2^2(p_0, p_1) = \inf_{p_t, v_t} \left\{ \int_0^1 \int_\Omega \|v_t(\mathbf{x})\|^2 p_t(\mathbf{x}) \, \mathrm{d}\mathbf{x}\mathrm{d}t \Big| \partial_t p_t = -\nabla \cdot (p_t v_t), \ p_{t=0} = p_0, p_{t=1} = p_1 \right\} \tag{A-15}$$

*and the optimal $v_t^*(\mathbf{x})$ can be expressed by Monge map between $p_0$ and $p_1$. i.e.,*

$$v_t(\mathbf{x} + t(T^*(\mathbf{x}) - \mathbf{x})) = T^*(\mathbf{x}) - \mathbf{x}, \ \mathbf{x} \sim p_0 \tag{A-16}$$

### A.2   Proof of Proposition 1

For the convenience of reading, here we restate Proposition 1 in the main text as Proposition A-2.

**Proposition A-2.** *Assume $c(\mathbf{x}_0, \mathbf{x}_1) = \|\mathbf{x}_0 - \mathbf{x}_1\|^2$ and if we enforce $\mathrm{P}_\#^0 \pi$ and $p_0$ to share the same support for admissible solution $\pi$ to problem (4), then we have $\min_\pi \mathcal{J}_{\mathrm{sot}}(\pi) = \min_{v_t, g_t} \mathcal{J}_{\mathrm{tpt}}^\lambda(v_t, g_t), \forall \lambda \in (0, 1)$. Moreover, for any $\lambda \in (0, 1)$, given the optimal transport plan $\pi^*$ to problem (4), let $p_\lambda^* \triangleq \mathrm{P}_\#^0 \pi^*$, then $\pi^*$ can be expressed as $\pi^* = (\mathrm{Id}, T^*)_\# p_\lambda^*$ where $T^*$ is the Monge map between $p_\lambda^*$ and $p_1$. Meanwhile, there exist a $g_t^*$ such that $p_\lambda^* = p_0(\mathbf{x})e^{\int_0^\lambda g_t^*(\mathbf{x})\mathrm{d}t}$, and a $v_t^*$ given by*

$$v_t^* \left( \mathbf{x} + \frac{t - \lambda}{1 - \lambda}(T^*(\mathbf{x}) - \mathbf{x}) \right) = \frac{T^*(\mathbf{x}) - \mathbf{x}}{1 - \lambda}, \tag{A-17}$$

*satisfying $(v_t^*, g_t^*) \in \arg\min_{v_t, g_t} \mathcal{J}_{\mathrm{tpt}}^\lambda(v_t, g_t)$.*

*Proof.* Recall that when $c(\mathbf{x}_0, \mathbf{x}_1) = \|\mathbf{x}_0 - \mathbf{x}_1\|^2$, problems (4) and (5) are respectively

$$\min_{\pi \geq 0} \mathcal{J}_{\mathrm{sot}}(\pi) \triangleq \int_{\Omega^2} \|\mathbf{x}_0 - \mathbf{x}_1\|^2 \, \mathrm{d}\pi(\mathbf{x}_0, \mathbf{x}_1) + \mathrm{KL}(\mathrm{P}_\#^0 \pi \| p_0) \quad \text{subject to} \quad \mathrm{P}_\#^1 \pi = p_1,$$

and

$$\min_{(v_t, g_t) \in \mathcal{C}_\lambda(p_0, p_1)} \mathcal{J}_{\mathrm{tpt}}^\lambda(v_t, g_t) \triangleq (1 - \lambda) \int_\Omega \int_\lambda^1 p_t(\mathbf{x}) \|v_t(\mathbf{x})\|^2 \mathrm{d}t\mathrm{d}x + \mathcal{H}(v_t, g_t, p_t),$$

where $\mathcal{H}(v_t, g_t, p_t) = \int_\Omega p_0(\mathbf{x})(e^{\int_0^\lambda g_t(\mathbf{x})\mathrm{d}t}(\int_0^\lambda g_t(\mathbf{x})\mathrm{d}t - 1) + 1)\mathrm{d}x, \mathcal{C}_\lambda(p_0, p_1) = \{(v_t, g_t) : \partial_t p_t = g_t p_t, t \in [0, \lambda]; \partial_t p_t = -\nabla \cdot (p_t v_t), t \in (\lambda, 1]\}$

**(1)** We first prove $\min_\pi \mathcal{J}_{\mathrm{sot}}(\pi) \geq \min_{v_t, g_t} \mathcal{J}_{\mathrm{tpt}}^\lambda(v_t, g_t)$.

Let $\pi^*$ be the optimal solution to problem (4). For any $\lambda \in (0,1)$, define $p_\lambda = P^0_{\#}\pi^*$, and let $w_\lambda(\mathbf{x}) := \frac{p_\lambda(\mathbf{x})}{p_0(\mathbf{x})}$. Define the growth as

$$\frac{\mathrm{d}}{\mathrm{d}t}\log w_t(\mathbf{x}) = g_t(\mathbf{x}), t \in [0,\lambda].$$

This implies the continuity equation $\partial_t p_t = g_t p_t$ for $t \in [0,\lambda]$, and hence we have $p_0(\mathbf{x})\exp\left(\int_0^\lambda g_t(\mathbf{x})dt\right) = P^0_{\#}\pi^*$.

By Brenier's theorem [60], we have $\pi^* = (\mathrm{Id}, T^*)_{\#}p_\lambda$, where $T^*(\mathbf{x})$ is the Monge map from $p_\lambda$ to $p_1$. Define

$$g_t(\mathbf{x}) = \frac{\log p_\lambda(\mathbf{x}) - \log p_0(\mathbf{x})}{\lambda}, \quad v_t\left(\mathbf{x} + \frac{t-\lambda}{1-\lambda}(T^*(\mathbf{x}) - \mathbf{x})\right) = \frac{T^*(\mathbf{x}) - \mathbf{x}}{1-\lambda}. \qquad \text{(A-18)}$$

Plugging into the dynamic objective and using the definition of KL-divergence:

$$\mathrm{KL}(p_\lambda \| p_0) = \int_\Omega p_0(\mathbf{x})\left(\frac{p_\lambda(\mathbf{x})}{p_0(\mathbf{x})}\left(\log\frac{p_\lambda(\mathbf{x})}{p_0(\mathbf{x})} - 1\right) + 1\right)\mathrm{d}\mathbf{x},$$

we obtain

$$\mathcal{J}^\lambda_{\mathrm{tpt}}(v_t, g_t) = \int_\Omega p_0(\mathbf{x})\left(e^{\int_0^\lambda g_t(\mathbf{x})\mathrm{d}t}\left(\int_0^\lambda g_t(\mathbf{x})\mathrm{d}t - 1\right) + 1\right)\mathrm{d}\mathbf{x} + (1-\lambda)\int_\Omega \|T^*(\mathbf{x}) - \mathbf{x}\|^2 p_\lambda(\mathbf{x})\mathrm{d}\mathbf{x}$$

$$= \mathrm{KL}(p_\lambda \| p_0) + \int_{\Omega^2} \|\mathbf{x}_0 - \mathbf{x}_1\|^2 \mathrm{d}\pi^*(\mathbf{x}_0, \mathbf{x}_1)$$

$$= \mathcal{J}_{\mathrm{sot}}(\pi^*)$$

Thus, $\mathcal{J}_{\mathrm{sot}}(\pi^*) = \mathcal{J}^\lambda_{\mathrm{tpt}}(v_t, g_t) \geq \min_{v_t, g_t} \mathcal{J}^\lambda_{\mathrm{tpt}}(v_t, g_t)$.

**(2)** To show the reverse inequality $\min_\pi \mathcal{J}_{\mathrm{sot}}(\pi) \leq \min_{v_t, g_t} \mathcal{J}^\lambda_{\mathrm{tpt}}(v_t, g_t)$, we assume the contrary that $\min_\pi \mathcal{J}_{\mathrm{sot}}(\pi) > \min_{v_t, g_t} \mathcal{J}^\lambda_{\mathrm{tpt}}(v_t, g_t)$. Let $(v_t^*, g_t^*)$ be an optimal solution to problem (5), and define

$$\tilde{p}_\lambda(\mathbf{x}) := p_0(\mathbf{x})\exp\left(\int_0^\lambda g_t^*(\mathbf{x})\mathrm{d}t\right),$$

Since the second-stage evolution $p_t$, for $t \in (\lambda, 1]$, is governed solely by the velocity field $v_t$ and does not involve mass creation or destruction, both $\tilde{p}_\lambda$ and $p_1$ have the same total mass. Thus, the Monge map $\tilde{T}^*$ from $\tilde{p}_\lambda$ to $p_1$ under quadratic cost is well-defined. Then, by the Benamou–Brenier formulation [71, 72], we have

$$v_t^*\left(\mathbf{x} + \frac{t-\lambda}{1-\lambda}(\tilde{T}^*(\mathbf{x}) - \mathbf{x})\right) = \frac{\tilde{T}^*(\mathbf{x}) - \mathbf{x}}{1-\lambda},$$

and the corresponding coupling $\tilde{\pi}^* := (\mathrm{Id}, \tilde{T}^*)_{\#}\tilde{p}_\lambda$ satisfies

$$\mathcal{J}_{\mathrm{sot}}(\tilde{\pi}^*) = \mathrm{KL}(\tilde{p}_\lambda \| p_0) + \int_{\Omega^2} \|\mathbf{x}_0 - \mathbf{x}_1\|^2 \mathrm{d}\tilde{\pi}^*(\mathbf{x}_0, \mathbf{x}_1)$$

$$= \int_\Omega p_0(\mathbf{x})\left(e^{\int_0^\lambda g_t^*(\mathbf{x})\mathrm{d}t}\left(\int_0^\lambda g_t^*(\mathbf{x})\mathrm{d}t - 1\right) + 1\right)\mathrm{d}\mathbf{x} + (1-\lambda)\int_\Omega \|v_t^*(\mathbf{x})\|^2 \tilde{p}_\lambda(\mathbf{x})\mathrm{d}\mathbf{x}$$

$$= \mathcal{J}^\lambda_{\mathrm{tpt}}(v_t^*, g_t^*) = \min_{v_t, g_t} \mathcal{J}^\lambda_{\mathrm{tpt}}(v_t, g_t) < \min_\pi \mathcal{J}_{\mathrm{sot}}(\pi)$$

which leads to the contradiction.

Combining (1) and (2), we have $\mathcal{J}_{\mathrm{sot}}(\pi^*) = \mathcal{J}^\lambda_{\mathrm{tpt}}(v_t^*, g_t^*)$.

Given $\pi^*$, we construct $g_t^*, v_t^*$ as in Eq. (A-18). According to the proof of (1), we have $\mathcal{J}^\lambda_{\mathrm{tpt}}(v_t^*, g_t^*) = \mathcal{J}_{\mathrm{sot}}(\pi^*) = \min_{v_t, g_t} \mathcal{J}^\lambda_{\mathrm{tpt}}(v_t, g_t)$. $\qquad \square$

## A.3 Proof and Empirical Evidence of Theorem 1

For the convenience of reading, here we restate Theorem 1 in the main text as Theorem A-3.

**Theorem A-3.** *Gvien the initial distribution $p_0$, denote the ending distribution of the two-period dynamics*

$$\partial_t p_t = g_t p_t, t \in [0, \lambda]; \quad \partial_t p_t = -\nabla \cdot (p_t v_t), t \in (\lambda, 1], \tag{A-19}$$

*as $p_1$, and denote the ending distribution of the joint dynamics starting from $p_0$*

$$\partial_t \tilde{p}_t = -\nabla \cdot (\tilde{p}_t \tilde{v}_t) + \tilde{g}_t \tilde{p}_t, \quad t \in [0, 1], \quad \tilde{p}_0 = p_0, \tag{A-20}$$

*as $\tilde{p}_1$, then it holds that $\tilde{p}_1 = p_1$.*

*Proof.* Given $\mathbf{x}_0 \sim p_0$, consider the two systems below.

*System I (original two-period transport dynamics):*

$$\begin{cases} \dfrac{\mathrm{d}\mathbf{x}_t}{\mathrm{d}t} = v_t(\mathbf{x}_t) \cdot \mathbb{I}_{(\lambda, 1]}(t), \\ \dfrac{\mathrm{d}}{\mathrm{d}t} \log w_t(\mathbf{x}_t) = g_t(\mathbf{x}_t) \cdot \mathbb{I}_{[0, \lambda]}(t), \end{cases} \tag{A-21}$$

where $w_t$ is the time-dependent weight function and $\mathbb{I}_\Omega$ is the indicator function, for any function $f, f \cdot \mathbb{I}_\Omega(t) = \begin{cases} f, & \text{If } t \in \Omega, \\ 0, & \text{Otherwise.} \end{cases}$.

*System II (joint dynamics defined via reparameterization (6)):*

$$\begin{cases} \dfrac{\mathrm{d}\mathbf{x}_t}{\mathrm{d}t} = \tilde{v}_t(\mathbf{x}_t), \\ \dfrac{\mathrm{d}}{\mathrm{d}t} \log w_t(\mathbf{x}_t) = \tilde{g}_t(\mathbf{x}_t), \end{cases} \tag{A-22}$$

with $\tilde{w}_t$ being the time-dependent weight under joint dynamics. We first recall the definition $\tilde{v}_t(\mathbf{x}) = (1 - \lambda) \cdot v_{(1-\lambda)t+\lambda}(\mathbf{x}), \tilde{g}_t(\mathbf{x}) = \lambda \cdot g_{\lambda t}\left(\psi_{\tilde{v},t}^{-1}(\mathbf{x})\right)$. To prove Theorem 1, it suffices to show that given the same initialization $\mathbf{x}_0, w_0(\mathbf{x}_0)$, the final state of system I $(\mathbf{x}_1, w_1(\mathbf{x}_1))$ and II $(\tilde{\mathbf{x}}_1, \tilde{w}_1(\tilde{\mathbf{x}}_1))$ are identical, *i.e.*, $\tilde{\mathbf{x}}_1 = \mathbf{x}_1$ and $\tilde{w}_1(\tilde{\mathbf{x}}_1) = w_1(\mathbf{x}_1)$.

$$\begin{aligned} \tilde{\mathbf{x}}_1 &= \mathbf{x}_0 + \int_0^1 \tilde{v}_t(\mathbf{x}) \, \mathrm{d}t \\ &= \mathbf{x}_0 + \int_0^1 v_{(1-\lambda)t+\lambda}(\mathbf{x}) \cdot (1 - \lambda) \, \mathrm{d}t \\ &= \mathbf{x}_0 + \int_\lambda^1 v_s(\mathbf{x}) \, \mathrm{d}s \quad (\text{let } s = (1-\lambda)t + \lambda) \\ &= \mathbf{x}_1. \end{aligned}$$

Meanwhile,

$$\begin{aligned} \log w_1(\mathbf{x}_1) &= \log w_\lambda(\mathbf{x}_\lambda) = \log w_\lambda(\mathbf{x}_0) \\ &= \log w_0(\mathbf{x}_0) + \int_0^\lambda g_s(\mathbf{x}_0) \, \mathrm{d}s \\ &= \log w_0(\mathbf{x}_0) + \int_0^1 \lambda \cdot g_{\lambda t}(\mathbf{x}_0) \, \mathrm{d}t \quad (\text{let } t = \frac{1}{\lambda}s) \\ &= \log w_0(\mathbf{x}_0) + \int_0^1 \lambda \cdot g_{\lambda t}(\psi_{\tilde{v},t}^{-1}(\mathbf{x})) \, \mathrm{d}t \quad (\mathbf{x} = \mathbf{x}_0 + \int_0^t \tilde{v}_s(\mathbf{x})\mathrm{d}s) \\ &= \log w_0(\mathbf{x}_0) + \int_0^1 \tilde{g}_t(\mathbf{x}) \, \mathrm{d}t \\ &= \log \tilde{w}_1(\tilde{\mathbf{x}}_1). \end{aligned}$$

Hence, the final state and mass at $t = 1$ under both systems coincide. Applying the above analysis to all $\mathbf{x}_0 \sim p_0$ completes the proof. □

**Empirical evidence of Theorem 1.** We provide the following example as an evidence to Theorem 1. Set $p_0 = \mathcal{N}(2, 0.5)$, $v_t(\mathbf{x}) = 2t$, $g_t(\mathbf{x}) = -\log(\mathbf{x}+1) + t^3$, using Eq.(6), we have $\tilde{v}_t(\mathbf{x}) = (1-\lambda)v_{(1-\lambda)t+\lambda}(\mathbf{x})$, $\tilde{g}_t(\mathbf{x}) = \lambda g_{\lambda t}(\psi_{\tilde{v},t}^{-1}(\mathbf{x})) = \lambda g_{\lambda t}(\mathbf{x} - ((1-\lambda)t + \lambda)^2 + \lambda^2)$. By setting $\lambda = 0.4$ and using numerical solvers to solve the corresponding continuity equation, we validate our correctness of our theorem. As shown in Fig. A-5, the two dynamics ended in the same distribution.

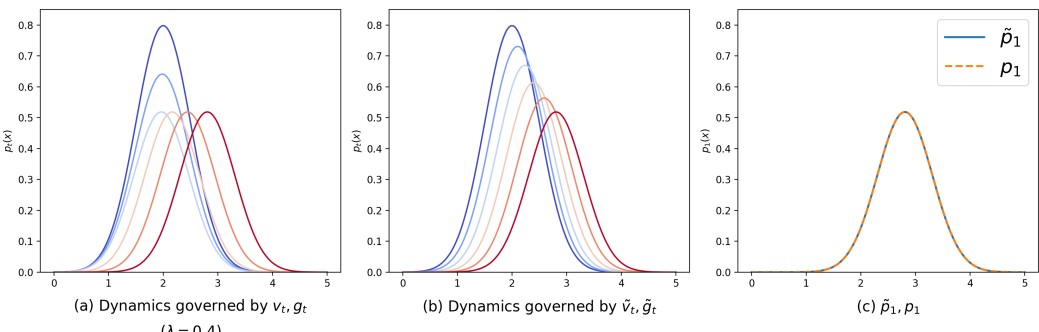

(a) Dynamics governed by $v_t, g_t$      (b) Dynamics governed by $\tilde{v}_t, \tilde{g}_t$      (c) $\tilde{p}_1, p_1$
($\lambda = 0.4$)

Figure A-5: Empirical evidence of Theorem 1.(a) Original two-period transport dynamics. (b) Joint dynamics defined via reparameterization 6. (c) The two dynamics ended in the same distribution.

## A.4 Motivation for time-independent growth function $\tilde{g}_t$ according to Eq. (9)

**Minimization of $L^2$ energy potential.** We aim to prove the following statement: Given a function $w_t(\mathbf{x})$ whose logarithmic derivative satisfies $\frac{\mathrm{d}}{\mathrm{d}t} \log w_t(\mathbf{x}) = g_t(\mathbf{x})$ and the boundary condition

$$\log w_1(\mathbf{x}) - \log w_0(\mathbf{x}) = \int_0^1 g_t(\mathbf{x})\, \mathrm{d}t,$$

then among all functions $g_t(\mathbf{x})$ satisfying this constraint, the one minimizing the energy functional

$$\mathcal{E}(g) := \int_\Omega \int_0^1 \|g_t(\mathbf{x})\|_2^2 \,\mathrm{d}t\mathrm{d}\mathbf{x}$$

is the constant function $\int_0^1 g_t(\mathbf{x})\mathrm{d}\mathbf{x} = \log w_1(\mathbf{x}) - \log w_0(\mathbf{x})$. To show this, we apply the method of Lagrange multipliers. Introduce a multiplier $\lambda(\mathbf{x})$ and consider the Lagrangian

$$\mathcal{L}(g_t) = \int_\Omega (\int_0^1 \left(g_t(\mathbf{x})^2 + \lambda(\mathbf{x})\, g_t(\mathbf{x})\right) \mathrm{d}t - \lambda(\mathbf{x})\left(\log w_1(\mathbf{x}) - \log w_0(\mathbf{x})\right))\mathrm{d}\mathbf{x}.$$

Taking the first variation of $\mathcal{L}$ with respect to $g_t(\mathbf{x})$ yields the optimality condition

$$\frac{\delta\mathcal{L}}{\delta g_t}(\mathbf{x}) = 2g_t(\mathbf{x}) + \lambda(\mathbf{x}) = 0 \;\Rightarrow\; g_t(\mathbf{x}) = -\frac{\lambda(\mathbf{x})}{2}.$$

This shows that the optimal $g_t(\mathbf{x})$ is constant with respect to time. Plugging this into the constraint gives $g_t(\mathbf{x}) = \log w_1(\mathbf{x}) - \log w_0(\mathbf{x})$. Hence, the constant function $g_t(\mathbf{x})$ is the unique minimizer of the energy functional under the given constraint.

**Explanation from the Malthusian growth model [58], *i.e.*, the exponential growth model.** The Malthusian growth model is $\frac{\mathrm{d}p_t}{\mathrm{d}t} = gp_t$, where $p_t$ is the population size and $g$ is a constant growth rate. Our model on the growth $\frac{\partial p_t}{\partial t} = g_t p_t$ is consistent with the above Malthusian growth model. In these models, $g$ is treated as a constant based on the assumption that the resources are abundant and the environment is stable, causing growth rates to remain relatively stable over time. In the context of scRNA-seq experiments, these conditions are typically satisfied because cells are often cultured or sampled under controlled laboratory conditions, where nutrient supply, temperature, and other environmental factors are maintained at constant levels. Thus, choosing this form is biologically reasonable, especially when there is no prior knowledge about the growth rate.

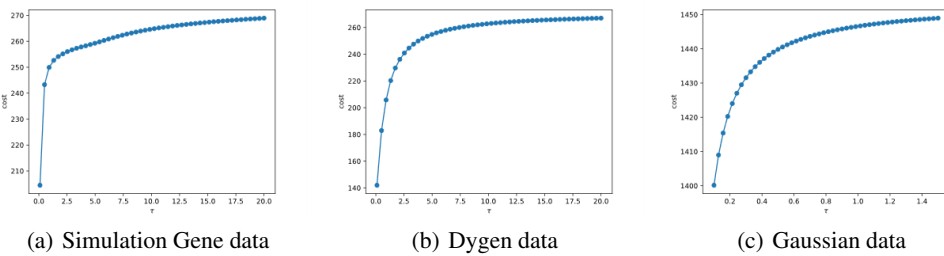

| (a) Simulation Gene data | (b) Dygen data | (c) Gaussian data |

Figure A-6: Fixing $\epsilon$, we plot the transport cost as a function of $\tau$. (a) In Simulation Gene, the curve flattens around $\tau = 5$; (b) In Dyngen, the transport cost also stabilizes near $\tau = 5$.

## B  Experimental Details

The model architecture has been described in Sect. 4. We next detail the strategy for selecting the hyperparameters $\epsilon$ and $\tau$ in Eq. (11), as well as provide a comprehensive description of each experiment. All experiments are performed on a single-core CPU without GPU acceleration, and all visualizations are based on projections onto the first two dimensions of the high-dimensional data.

### B.1  Determine the Entropy Regularization Parameter $\epsilon$ and Relaxation Coefficient $\tau$

Our algorithm involves solving a static semi-relaxed optimal transport problem (11), which includes two critical hyperparameters $\epsilon$ and $\tau$. These parameters play a pivotal role in the behavior and stability of the model. On one hand, if $\epsilon$ is too small, it may lead to numerical instability; if it is too large, it can cause incorrect cross-branch matching in unbalanced data scenarios. To select $\epsilon$, we first set $\tau$ to a moderately large value, such as $\tau = 50$, and then perform a grid search over increasing values of $\epsilon$ starting from 0 until numerical stability is achieved. The choice of $\tau$ is even more crucial. When $\tau$ is too small, the KL divergence term between $P^0_\# \pi$ and $p_0$ receives a weak penalty. As a result, the transport mass becomes overly concentrated around points close to $p_1$, leading to highly uneven marginals $\pi \mathbf{1}_m$ in the discrete setting. This causes instability in the learning of the growth function $g$ and may result in erroneous modeling. Conversely, as $\tau \to +\infty$, Eq. (11) reduces to a balanced OT problem, which contradicts the unbalanced nature of our formulation. To determine a suitable value for $\tau$, we fix the previously selected small $\epsilon$ and gradually increase $\tau$ while observing the variation in $\sum \pi_{ij} c_{ij}$. This curve typically exhibits an increasing-then-flattening trend. Similar to the "elbow metho" used in clustering to select the optimal number of clusters, we identify the region where the curve becomes stable and choose $\tau$ accordingly, as illustrated in Fig. A-6.

On the *simulation gene* dataset, we fix $\epsilon = 0.001$ and evaluate the model with different values of $\tau \in \{5, 10, 15, 20\}$, all of which lie within the identified stable region. The corresponding model performance metrics are summarized in Tab. A-4. We observe that the models trained with $\tau$ in this range are stable, validating the effectiveness of our hyperparameter selection strategy.

Table A-4: Fix $\epsilon = 0.001$, change different $\tau$ on simulation gene data, we present the Wasserstein-1 distance of each timepoint.

| Models | $t_1$ | | $t_2$ | | $t_3$ | | $t_4$ | |
|---|---|---|---|---|---|---|---|---|
| | $\mathcal{W}_1$ | RME | $\mathcal{W}_1$ | RME | $\mathcal{W}_1$ | RME | $\mathcal{W}_1$ | RME |
| VFGM ($\tau = 5$) | 0.046 | 0.007 | 0.062 | 0.001 | 0.053 | 0.003 | 0.063 | 0.003 |
| VFGM ($\tau = 10$) | 0.041 | 0.005 | 0.053 | 0.005 | 0.038 | 0.007 | 0.040 | 0.008 |
| VFGM ($\tau = 15$) | 0.045 | 0.003 | 0.056 | 0.003 | 0.046 | 0.005 | 0.052 | 0.004 |
| VFGM ($\tau = 20$) | 0.043 | 0.007 | 0.057 | 0.004 | 0.045 | 0.007 | 0.058 | 0.011 |

## B.2 Simulation of Gene Data

Following the setup of [26], the dynamics of the simulated gene regulatory network are governed by the following system of differential equations:

$$\frac{\mathrm{d}X_1}{\mathrm{d}t} = \frac{\alpha_1 X_1^2 + \beta}{1 + \alpha_1 X_1^2 + \gamma_2 X_2^2 + \gamma_3 X_3^2 + \beta} - \delta_1 X_1 + \eta_1 \xi_t,$$

$$\frac{\mathrm{d}X_2}{\mathrm{d}t} = \frac{\alpha_2 X_2^2 + \beta}{1 + \gamma_1 X_1^2 + \alpha_2 X_2^2 + \gamma_3 X_3^2 + \beta} - \delta_2 X_2 + \eta_2 \xi_t, \tag{A-23}$$

$$\frac{\mathrm{d}X_3}{\mathrm{d}t} = \frac{\alpha_3 X_3^2}{1 + \alpha_3 X_3^2} - \delta_3 X_3 + \eta_3 \xi_t,$$

where $X_i(t)$ denotes the expression level of gene $i$ at time $t$. Genes $X_1$ and $X_2$ mutually inhibit each other while self-activating, forming a toggle-switch regulatory motif. An external signal $\beta$ independently activates both $X_1$ and $X_2$, whereas $X_3$ inhibits the expression of both $X_1$ and $X_2$.

The parameters $\alpha_i$ and $\gamma_i$ determine the strengths of self-activation and inhibition, respectively. The $\delta_i$ terms denote degradation rates, and $\eta_i \xi_t$ represents additive Gaussian noise with intensity $\eta_i$. The cell division rate is positively correlated with $X_2$ expression and is calculated as

$$g = \frac{\alpha_2 X_2^2}{1 + X_2^2}\%.$$

At each cell division event, daughter cells inherit gene expression values $(X_1, X_2, X_3)$ from the parent, subject to small perturbations $\eta_d \mathcal{N}(0, 1)$ per gene. Post-division, cells evolve independently according to the same stochastic dynamics.

**Initial Conditions and Simulation Setup.** Initial expression states are sampled from two normal distributions: $\mathcal{N}([2, 0.2, 0], 0.01)$ and $\mathcal{N}([0, 0, 2], 0.01)$. At every step, negative values are clipped to zero. Gene expression data is recorded at time points $t \in \{0, 8, 16, 24, 32\}$.

By setting $\tau = 10$ and $\epsilon = 0.003$ according to our selection scheme described in Appendix B.1 and illustrated in Fig. A-6 (a). Our model achieve the best performance in terms of (weighted) $\mathcal{W}_1$ and relative mass error as shown in Tab. 1.

Table A-5: Simulation parameters on gene regulatory network [26].

| Parameter | Value | Description |
|---|---|---|
| $\alpha_1$ | 0.5 | Strength of self-activation for $X_1$ |
| $\gamma_1$ | 0.5 | Inhibition of $X_2$ by $X_1$ |
| $\alpha_2$ | 1 | Strength of self-activation for $X_2$ |
| $\gamma_2$ | 1 | Inhibition of $X_1$ by $X_2$ |
| $\alpha_3$ | 1 | Strength of self-activation for $X_3$ |
| $\gamma_3$ | 10 | Half-saturation constant for inhibition |
| $\delta_1$ | 0.4 | Degradation rate for $X_1$ |
| $\delta_2$ | 0.4 | Degradation rate for $X_2$ |
| $\delta_3$ | 0.4 | Degradation rate for $X_3$ |
| $\eta_1$ | 0.05 | Noise intensity for $X_1$ |
| $\eta_2$ | 0.05 | Noise intensity for $X_2$ |
| $\eta_3$ | 0.05 | Noise intensity for $X_3$ |
| $\eta_d$ | 0.014 | Noise for perturbation during cell division |
| $\beta$ | 1 | External activation signal |
| $\mathrm{d}t$ | 1 | Simulation time step |
| Time Points | $\{0, 8, 16, 24, 32\}$ | Observation time points |

**Growth rate correlation analysis.**

To validate the accuracy of our growth rate modeling and learning, we perform correlation analysis between the predicted growth rates and the ground truth. We utilize this dataset, which is generated

from the prescribed dynamical systems (Eq. (A-23)). Since the ground truth growth rates can be computed analytically, we conduct **correlation analysis on out-of-distribution (OOD) time points**—excluded from the training data—to assess the generalization capability of VGFM. The results are summarized in Table A-6.

Table A-6: Correlation analysis on four OOD time points for Simulation Gene Data

| OOD Time Point | Pearson Correlation Coefficient |
|---|---|
| $t_{0.5}$ | 0.980 |
| $t_{1.5}$ | 0.992 |
| $t_{2.5}$ | 0.995 |
| $t_{3.5}$ | 0.996 |

The growth rates predicted by our VGFM exhibit strong correlation with the ground truth values.

**Uncertainty quantification.** Although our current model is deterministic, we incorporate dropout into the velocity and growth networks to quantify predictive uncertainty. During inference, we compute the average variance for each dimension of velocity and growth across all data points, considering different branches (0 denotes the quiescent region *i.e.*, the lower left corner of Fig. 3 while 1 denotes the region where exhibits mass variation and state transition) and time points. We conduct the experiment with a dropout rate of 0.1, performing 5 stochastic forward passes through the velocity and growth networks for all data points. The results are presented in Table A-7.

Table A-7: Uncertainty estimation of predicted velocity (2-dimensional) and growth on Simulation Gene Data. $\text{var}_{x_1}$ and $\text{var}_{x_2}$ represent the average variance of the first and second velocity dimensions, respectively, while $\text{var}_g$ denotes the variance of growth. All variance values are scaled by $10^{-3}$.

| Branch | Time | $\text{var}_{x_1}$ ($\times 10^{-3}$) | $\text{var}_{x_2}$ ($\times 10^{-3}$) | $\text{var}_g$ ($\times 10^{-3}$) |
|---|---|---|---|---|
| 0 | 0 | 0.1080 | 0.1000 | 0.0550 |
| 0 | 1 | 0.0770 | 0.0600 | 0.0660 |
| 0 | 2 | 0.0660 | 0.0670 | 0.1100 |
| 0 | 3 | 0.0850 | 0.0820 | 0.1910 |
| 0 | 4 | 0.1220 | 0.1370 | 0.3500 |
| 1 | 0 | 2.9320 | 6.5010 | 0.2510 |
| 1 | 1 | 3.0230 | 4.2870 | 0.3060 |
| 1 | 2 | 2.0410 | 1.9780 | 0.5590 |
| 1 | 3 | 0.9530 | 0.9760 | 0.7660 |
| 1 | 4 | 0.2800 | 0.3100 | 1.1680 |

We analyze the uncertainty patterns from both spatial and temporal perspectives:

For spatial analysis, cells in branch 0 exhibit minimal movement with little variation in state and counts. Correspondingly, the variances of $v_1$, $v_2$, and $g$ in this branch are consistently small, indicating low predictive uncertainty. In contrast, branch 1 demonstrates substantially larger variances, reflecting higher uncertainty in predictions.

For temporal analysis, The Simulation Gene Data features significant state transitions and quantity changes during initial phases, with diminishing magnitudes over time. The variance patterns of $v_1$, $v_2$, and $g$ accurately capture this temporal evolution, validating our uncertainty estimation approach. Specifically, for branch 1, the of $v$ decrease monotonically over time, aligning with the reduced dynamics in later stages.

**Effect of sampling bias.** In cellular dynamics reconstruction, it is conventionally assumed that observed datasets accurately capture the temporal evolution of the true cell distribution. Consequently, the observed cell counts should align with the biological processes of cell proliferation and death. Evaluating the robustness of VGFM when trained on biased observational data holds significant practical relevance.

To generate non-uniform or subsampled data, we resample the original dataset with perturbed branch ratios. Specifically, for each time point $t$, we randomly sample a perturbation factor $a_t \sim \mathcal{N}(0, \sigma^2)$. Given an original branch ratio of $s_t : (1-s_t)$, the resampled ratio becomes $(s_t - |a_t|) : (1 - s_t + |a_t|)$,

where $\sigma$ quantifies the bias level. We train our model on the resampled data and evaluate its performance on the original dataset to assess VGFM's robustness.

Table A-8: Performance evaluation ($\mathcal{W}_1$/RME metrics) on Simulation Gene Data with varying sampling bias levels, averaged over three random seeds.

| Bias Factor $\sigma$ | $t_1$ | $t_2$ | $t_3$ | $t_4$ |
|---|---|---|---|---|
| 0 (unbiased) | 0.041/0.007 | 0.053/0.005 | 0.038/0.007 | 0.040/0.008 |
| 0.1 | 0.074/0.035 | 0.094/0.051 | 0.075/0.063 | 0.077/0.032 |
| 0.2 | 0.085/0.059 | 0.113/0.073 | 0.134/0.062 | 0.123/0.125 |

### B.3 Dyngen Data

Dyngen is a multi-modal simulation engine for studying dynamic cellular processes at single-cell resolution [62]. We follow the setup of [31], which simulates gene expression time-series data that mimics cell proliferation processes, including branching and temporal progression. In the analyzed instance of the dyngen dataset, the samples span five discrete time points, labeled from 0 to 4, capturing the temporal evolution of the cell population.

We use PHATE [63] to reduce its dimensions to 5, the same as [31]. Starting from time point 0, cells can be divided into two branches based on the sign of the second PHATE coordinate ($x_2$). Quantitative analysis reveals a pronounced branch imbalance: for instance, at time point 4, 88 cells belong to one branch while 213 belong to the other, which poses a great challenge to the models that do not consider unbalancedness, resulting in cross-branch inference. Throughout this study, we assume that the temporal change in cell counts within each branch is entirely governed by a growth function, without contributions from migration or observational noise.

By setting $\tau = 5$ and $\epsilon = 0.03$ according to our selection scheme described in Appendix B.1 and illustrated in Fig. A-6 (b). Our model successfully avoids cross-branch reconstructions in trajectory modeling, as illustrated in Fig. A-7. This leads to generated samples that remain well-aligned with the underlying manifold. In terms of growth rate estimation, our model also accurately captures the rapid expansion observed in the branch below between time point 3 and time point 4, where the number of observed cells increases dramatically from 25 to 213, indicating the highest growth rate in the system.

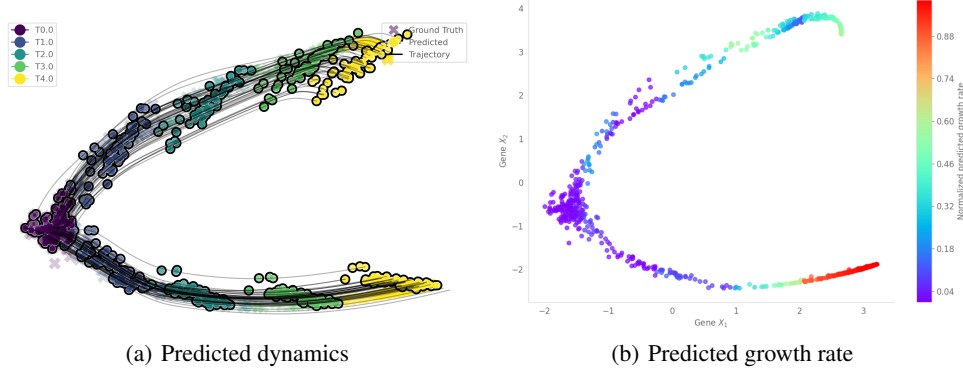

(a) Predicted dynamics        (b) Predicted growth rate

Figure A-7: Predicted dynamics and growth rates by VGFM on Dyngen data.

### B.4 Gausssian 1000D Data

DeepRUOT [26] employs a high-dimensional Gaussian mixture distribution (100D) [64] to evaluate the scalability of models. In this work, we adopt an even more challenging setting by testing scalability at 1000 dimensions. Following the setup of [26], for the initial distribution, we generated 400 samples from the Gaussian located lower in the $(x_1, x_2)$ plane, and 100 samples from the Gaussian positioned higher. For the final distribution, we generated 1,000 samples from the upper Gaussian, and 200

samples each from the two lower Gaussians, and assume cells in the upper region exhibit proliferation without transport [26]. We observe that simulation-based methods [26] encounter training instability in the 1000-dimensional setting and flow matching-based methods fail to learn a reliable velocity field due to the unbalancedness of data, as shown in Tab. 1. (Some works incorporating unbalancedness [37, 42] may as well reconstruct correct velocity field but fail to construct growth rate). In contrast, By setting $\tau = 5$ and $\epsilon = 0.03$ according to our selection scheme described in Appendix B.1 and illustrated in Fig. A-6 (c). Our method remains effective even in high-dimensional regimes for both velocity and growth field learning.

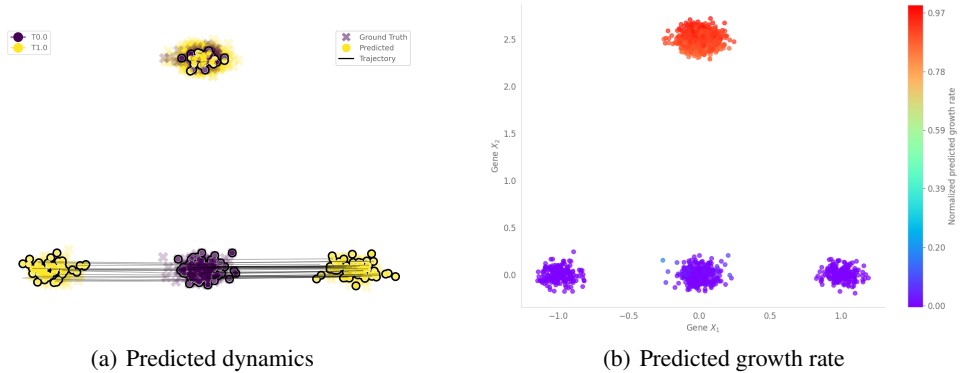

(a) Predicted dynamics        (b) Predicted growth rate

Figure A-8: Predicted dynamics and growth rates by VGFM on Gaussian 1000D data.

## B.5 Additional Experiment: Mouse Hematopoiesis Data

We also validate our algorithm on the mouse hematopoiesis data previously analyzed in [23, 26, 73]. This dataset leverages lineage tracing to track differentiation trajectories. After applying batch correction to integrate data across multiple experiments, the cells were embedded into a two-dimensional force-directed layout (SPRING plot). The resulting visualization reveals a pronounced bifurcation, where early progenitor cells diverge into two distinct differentiation lineages.

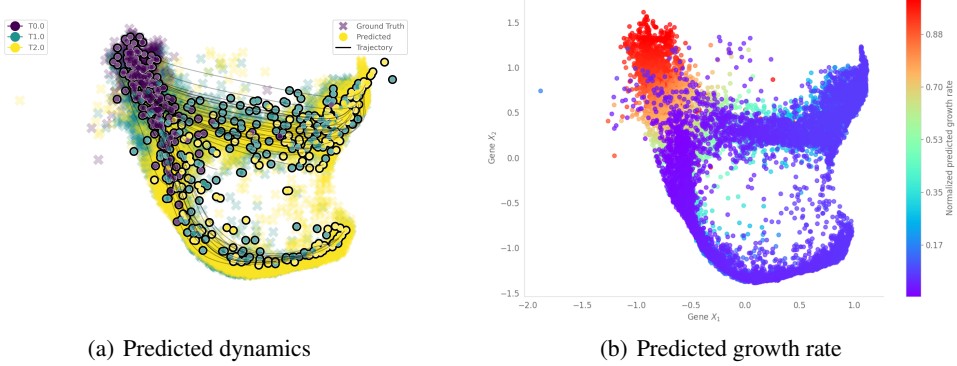

(a) Predicted dynamics        (b) Predicted growth rate

Figure A-9: Predicted dynamics and growth rates by VGFM on mouse hematopoiesis data.

Our model effectively captures the underlying branching structure in the data, and the predicted growth rates align well with those reported in [26], as well as with established biological priors and are consistent with known biological lineage patterns. Following the evaluation protocol of [26], we also compute the Wasserstein-1 distance as a quantitative metric. Under this evaluation, by setting $\tau = 20$ and $\epsilon = 0.005$ according to our selection scheme described in Appendix B.1, we train the model with a warm-up stage of 200 iterations. After that, the distribution fitting loss $\mathcal{L}_{\mathrm{OT}}$ is introduced for an additional 100 training epochs. Our method achieves superior performance, demonstrating improved trajectory inference. In addition, we evaluate the RME (Relative Mass Error)

metric introduced in this paper. The results indicate that our model significantly outperforms the other baselines in terms of mass-matching reconstruction accuracy, as shown in Tab. A-9 and Fig. A-10.

Table A-9: $\mathcal{W}_1$ and RME over all time on mouse hematopoiesis data. Part of the results were adopted from [26].

| Models | $t_1$ | | $t_2$ | |
|---|---|---|---|---|
| | $\mathcal{W}_1$ | RME | $\mathcal{W}_1$ | RME |
| SF$^2$M [41] | 0.167 | — | 0.190 | — |
| uAM[44] | 0.745 | — | 0.777 | — |
| UDSB [57] | 0.388 | 0.159 | 0.128 | 0.249 |
| TIGON [23] | 0.314 | 0.124 | 0.342 | 0.177 |
| DeepRUOT [26] | 0.145 | 0.140 | 0.132 | 0.202 |
| VGFM | **0.115** | **0.043** | **0.094** | **0.019** |

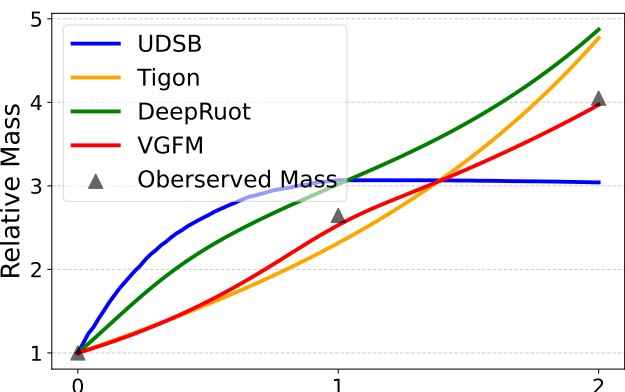

Figure A-10: Comparison of predicted relative mass (UDSB, TIGON, DeepRUOT, VGFM) with observed values from mouse hematopoiesis data.

## B.6   EB Data

Extensive experiments on the Embryoid Body (EB) [65] dataset with varying PCA dimensions are conducted in Sect. 4. We preprocess this dataset first by normalizing EB (5D) and leaving EB (50D) unnormalized. The proposed VGFM demonstrates strong performance in both cellular dynamics reconstruction and growth prediction, as illustrated in Fig. A-11. Specifically, we set the parameters as $\epsilon = 0.01$, $\tau = 5$ and normalize the cost matrix to ensure numerical stability when computing the semi-relaxed optimal transport plans (Eq. (11)).

**Big batch strategy for large-scale datasets.** When the number of observed data points is large, computing the transport plan between two consecutive time points using Eq. (11) becomes computationally inefficient. To address this, we adopt a Big-Batch Strategy. Specifically, we partition the data at each time point into $n$ subsets, referred to as Big Batches. For each of the $n$ Big Batches, we precompute and store the transport plans between adjacent time points. During training, we randomly select one Big Batch and sample smaller mini-batches from it based on the precomputed transport plan for model optimization. In real-world datasets, we set $n = 5$, while in synthetic datasets above we do not apply this strategy, *i.e.*, $n = 1$, as the sample size is relatively smaller. The mini-batch size is set to 256 for all experiments, except for Dyngen, where we use a smaller batch size of 60 due to its limited number of samples.

**Comparison with sinkhorn divergence as distribution fitting loss $\mathcal{L}_{\text{OT}}$.** We compute the distribution fitting loss $\mathcal{W}_1$ in Sect. 3.4 using EMD distance by `pot` library, following MIOFlow [31] and DeepRUOT [26]. Particularly, we compute the OT using the `pot` library, using the function

`pot.emd()`. This function solves for $\pi$ using a network flow algorithm, which is not compatible with PyTorch's automatic differentiation. However, the gradients of the cost function $c$ can still be backpropagated.

We replace the EMD distance with Sinkhorn divergence [70], which is fully differentiable and enjoys many favorable properties, using CUDA/C++ based `geomloss` library. Tab. A-10 verified the effectiveness of Sinkhorn divergence for the distribution fitting loss.

Sinkhorn divergence between distributions $p_0$ and $p_1$ is defined as $S_\epsilon(p_0, p_1) = \mathcal{W}_\epsilon(p_0, p_1) - \frac{1}{2}\mathcal{W}_\epsilon(p_0, p_1) - \frac{1}{2}\mathcal{W}_\epsilon(p_1, p_1)$, where $\mathcal{W}_\epsilon(p_0, p_1)$ is the entropy-regularized optimal transport distance (Sinkhorn distance) and $\epsilon > 0$ is the entropy regularization parameter.

Table A-10: Comparison of EMD and sinkhorn divergence (with different entropic regularization parameters) on EB 50D dataset with metrics ($\mathcal{W}_\infty$/RME at each timepoint).

| $\mathcal{L}_{\mathrm{OT}}$ | $t_1$ | $t_2$ | $t_3$ | $t_4$ | Training time(min) |
|---|---|---|---|---|---|
| EMD | 7.951/0.039 | 8.747/0.042 | 9.244/0.019 | 9.620/0.044 | 13 |
| $S_\epsilon(\epsilon = 0.001)$ | 7.902/0.018 | 8.767/0.013 | 9.063/0.083 | 9.507/0.096 | 9 |
| $S_\epsilon(\epsilon = 0.002)$ | 7.904/0.032 | 8.791/0.033 | 9.111/0.104 | 9.523/0.120 | 9 |
| $S_\epsilon(\epsilon = 0.005)$ | 7.917/0.048 | 8.768/0.062 | 9.086/0.137 | 9.518/0.151 | 9 |
| $S_\epsilon(\epsilon = 0.01)$ | 7.904/0.035 | 8.801/0.038 | 9.102/0.113 | 9.511/0.127 | 9 |
| $S_\epsilon(\epsilon = 0.05)$ | 7.908/0.036 | 8.787/0.045 | 9.080/0.120 | 9.512/0.129 | 9 |
| $S_\epsilon(\epsilon = 0.1)$ | 7.916/0.030 | 8.773/0.022 | 9.111/0.091 | 9.591/0.108 | 9 |

It can be found that the sinkhorn divergence achieves further improvement on the accuracy of dynamics reconstruction, and costs less time for computing the distribution fitting loss.

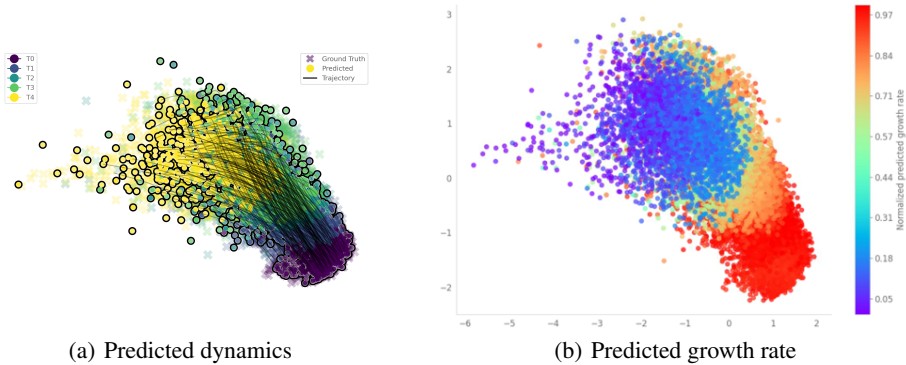

(a) Predicted dynamics  (b) Predicted growth rate

Figure A-11: Predicted dynamics and predicted growth rates by VGFM on 5D EB data with timepoint 1 held out.

We also examine whether the distribution fitting loss might lead to overfitting on the given population snapshots. In the hold-one-out experiments (Tab. 2), we relabel the timestamps of the four timepoints as 0, 1, 2, 3, and 4. Following the protocol in [38], we evaluate the model by holding out one intermediate timepoint at a time (*i.e.*, 1, 2, and 3) and report the average Wasserstein-1 distance. While hold-one-out experiments have already validated the superiority of VFGM over other methods, we further analyze the loss curves of both the distribution fitting loss and the $\mathcal{W}_1$ distance on a hold-out distribution (*e.g.*, the distribution at time 1) that is not seen during training (Fig. A-12). The results show that the $\mathcal{W}_1$ distance on the hold-out distribution decreases during the optimization of $\mathcal{L}_{\mathrm{OT}}$, indicating the absence of overfitting and demonstrating the generalization ability of the distribution fitting loss in modeling unseen distributions.

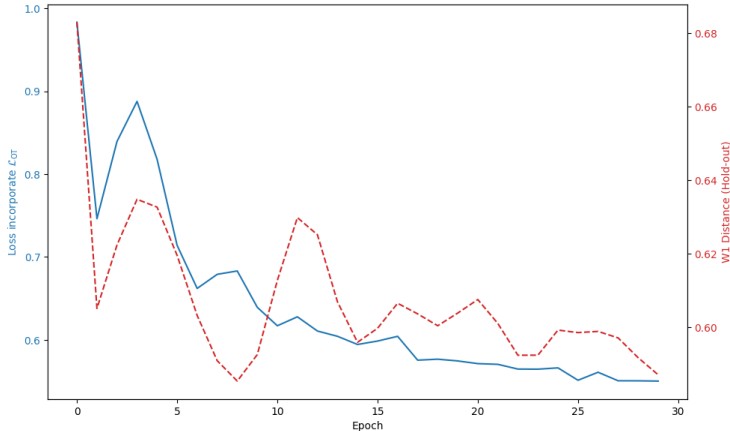

Figure A-12: Loss curves of the training loss after incorporating $\mathcal{L}_{\mathrm{OT}}$ (blue) and the $\mathcal{W}_1$ distance on a hold-out distribution at timepoint 1 (red) on 5D EB data.

## B.7 CITE Data

CITE-seq (Cellular Indexing of Transcriptomes and Epitopes by Sequencing) [66] is an advanced technique that enables the simultaneous profiling of transcriptomes and surface protein expression at the single-cell level through the use of antibody-derived tags. In this study, we utilize only the gene expression matrix from the CITE-seq dataset, and preprocess the data by normalizing CITE (5D) and leaving CITE (50D) unnormalized. The experimental settings for the CITE dataset are consistent with those used in the EB data experiments, employing PCA with 5 and 50 dimensions, and hyperparameters set as $\epsilon = 0.01$, $\tau = 5$. We relabel the timestamps of the four timepoints as 0, 1, 2, and 3. Following the protocol in [38], we evaluate the model by holding out one intermediate timepoint at a time (*i.e.*, 1 and 2) and report the average Wasserstein-1 distance.

To assess the model's capability to capture state trajectories (Fig. A-13 (a)) and predict mass growth (Fig. A-13 (b)), the distribution at time point 1 is held out. Additionally, the distribution at time point 2 is held out to evaluate the performance of the $\mathcal{L}_{\mathrm{OT}}$ (Fig. A-14). The strong performance observed in the visualizations substantiates the potential of the proposed VGFM framework.

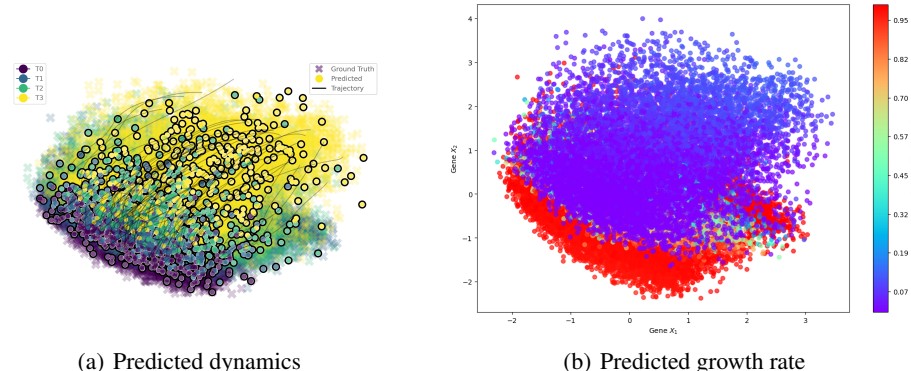

(a) Predicted dynamics          (b) Predicted growth rate

Figure A-13: Predicted dynamics and predicted growth rates by VGFM on 5D CITE Data with timepoint 1 held out.

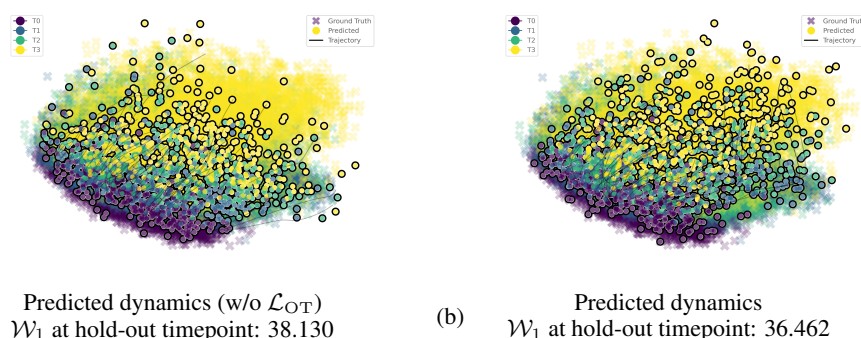

(a)     Predicted dynamics (w/o $\mathcal{L}_{\mathrm{OT}}$)
$\mathcal{W}_1$ at hold-out timepoint: 38.130

(b)     Predicted dynamics
$\mathcal{W}_1$ at hold-out timepoint: 36.462

Figure A-14: Visualization of predicted dynamics by (a) VFGM (w/o $\mathcal{L}_{\mathrm{OT}}$) and (b) VGFM on CITE (50D) dataset, where the hold-out time is the second intermediate timepoint.

## B.8    Pancreas Data Analysis

To further explore the scalability of VGFM, we applied our model and compared methods explicitly modeling $g_t(x)$ [23, 26] to the Pancreas dataset [67] comprising measurements of the developing mouse pancreas. We select cells at day 14.5 and 15.5 and relabel them as 0, 1 and select 2000 highly variable genes. As shown in [37], we assume the following cell type transitions are exclusively correct (denoted by $\rightarrow$), *i.e.*, there is no descending cell type (or set of cell types) other than the given one. We partition all considered cell type transitions into three branches.

**Endocrine branch (ED) transitions.**

- Fev+ Alpha (**FA**) $\rightarrow$ Alpha (**A**)
- Fev+ Beta (**FB**) $\rightarrow$ Beta (**B**)
- Fev+ Delta (**FD**) $\rightarrow$ Delta (**D**)
- Fev+ Epsilon (**FE**) $\rightarrow$ Epsilon (**E**)
- A $\rightarrow$ A
- B $\rightarrow$ B
- D $\rightarrow$ D
- E $\rightarrow$ E

**Ngn3 EP transitions.**

- Ngn3 high early (**NE**) $\rightarrow$ ED
- Ngn3 high late (**NL**) $\rightarrow$ ED

**Non-endocrine branch (NEB) transitions.**

- Ductal (**DU**) $\rightarrow$ DU
- Tip (**T**) $\rightarrow$ Acinar (**AC**)
- AC $\rightarrow$ AC

We observed that VGFM is the only method showing a steadily decreasing training loss, both for $\mathcal{L}_{\mathrm{VGFM}}$ and $\mathcal{L}_{\mathrm{OT}}$. We report $\mathcal{W}_1$ and RME as shown in Tab. A-11.

This indicates that both loss terms are still effective in high-dimensional real-world data.

**Analysis on mean and variance.** We calculated the means and variances of the real and generated gene at day 15.5 and plotted the corresponding mean-variance trend (Fig. A-15 (a), (b)) and histograms (Fig. A-15 (c), (d)). The results show that the generated samples closely follow the mean–variance trend of the real data, especially by incorporating $\mathcal{L}_{\mathrm{OT}}$.

Table A-11: $\mathcal{W}_1$ and RME at day 15.5 between real data and generated data from VGFM and its variant.

| Method | $\mathcal{W}_1$ | RME |
|---|---|---|
| VGFM (w/o $\mathcal{L}_{\mathrm{OT}}$) | 27.002 | 0.025 |
| VGFM | 24.416 | 0.017 |

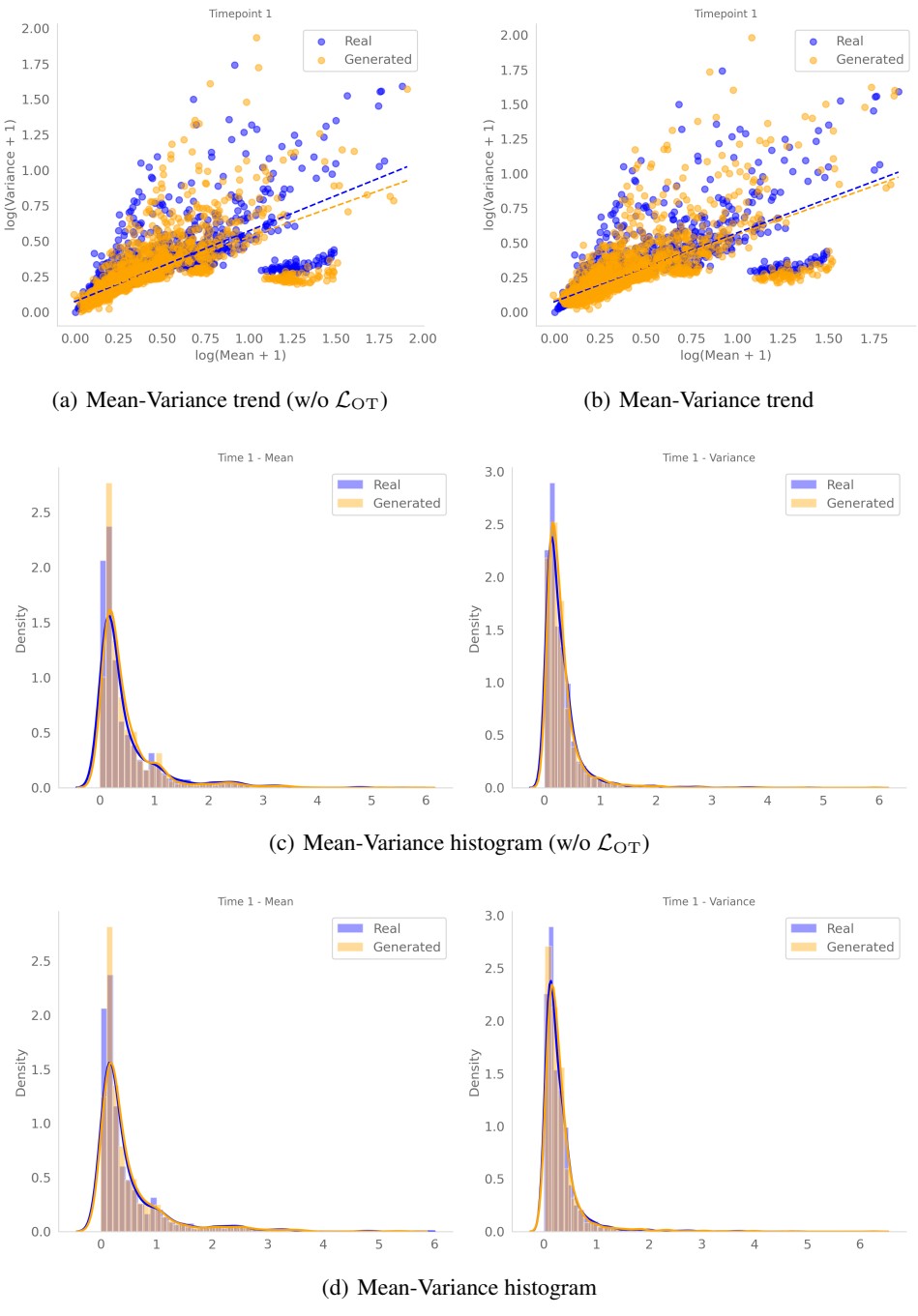

(a) Mean-Variance trend (w/o $\mathcal{L}_{\mathrm{OT}}$)

(b) Mean-Variance trend

(c) Mean-Variance histogram (w/o $\mathcal{L}_{\mathrm{OT}}$)

(d) Mean-Variance histogram

Figure A-15: mean-variance trend ((a), (b)) and histograms ((c), (d)) of real and generated gene data.

**Interpretable learned growth function.** We highlight that our main contribution lies in modeling and training the growth function $g_t(x)$, with VGFM in the flow matching framework, to leverage snapshot mass change for learning $g_w(x, t)$. During training, the growth loss rapidly converges to a negligible value, much faster than the velocity loss. To further investigate $g_w(x, t)$, We first calculate growth rate of each cell at the initial time point (day 14.5) and visualize them in Fig. A-16. At this time, the proliferation observed during this developmental stage mainly originates from Acinar and Ductal cells. Remarkably, our model successfully recapitulates this pattern without being given any prior knowledge of the cell types, demonstrating its ability to infer biologically meaningful dynamics directly from the data.

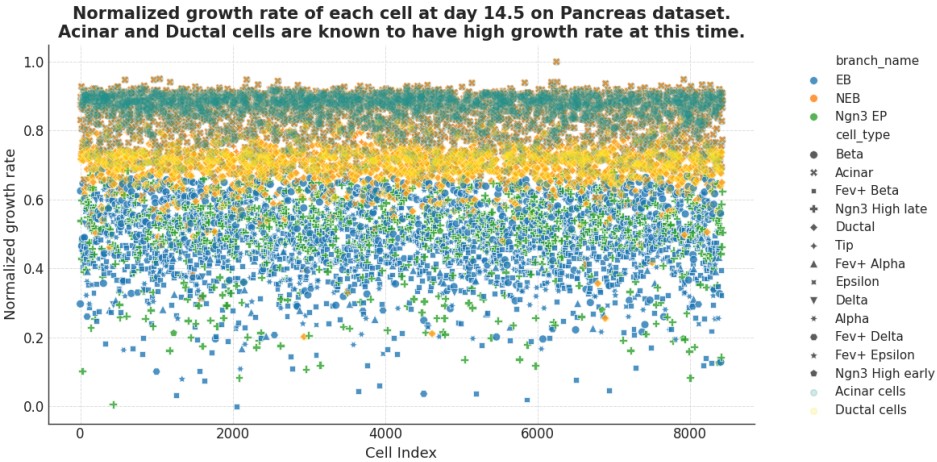

Figure A-16: Normalized growth rate of each cell at time point 0 (day 14.5).

We also demonstrate that the learned $g_w(x, t)$ not only successfully recapitulates the relative growth rates of different cell types without any prior information about cell identities, but also that the branch-wise mass obtained from numerical integration of the ODE closely matches the total branch mass at time point 1.

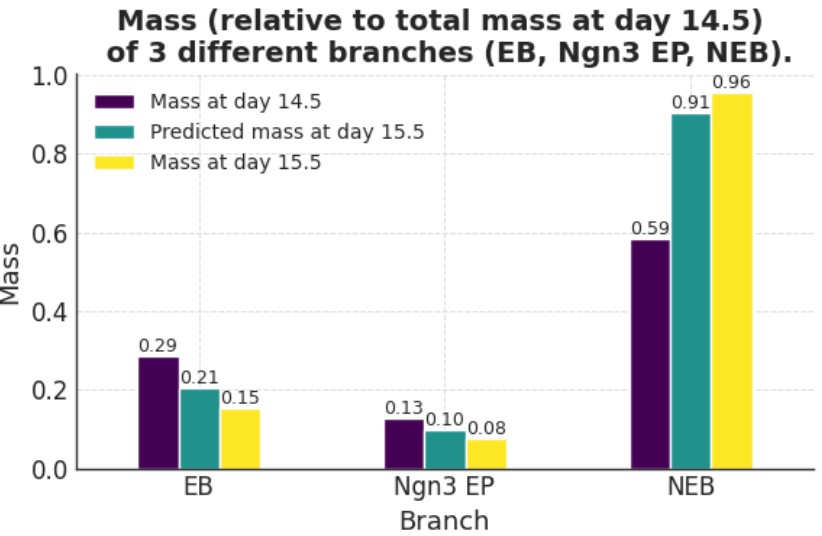

Figure A-17: Relative mass of different branches

we then follow [23], for each time $t$, compute the element-wise absolute value $\left|\frac{\partial g(x,t)}{\partial x_i}\right|$ in the gene space for every cell, and average across cells to quantify each gene's contribution to growth

dynamics. From our analysis of $g_w(x, t)$, We have identified Pnliprp1, Clps, and Ctrb1, as shown in Fig. A-18. These genes are well-known markers of Acinar cells and are highly expressed in the exocrine pancreas. Notably, as shown above, cells from the non-endocrine branch, particularly Acinar and Ductal cells, are much more abundant at later stages due to their high proliferation rates. This alignment between the learned growth-driving genes and known biological processes indicates that VGFM captures growth dynamics in a biologically interpretable manner, successfully linking the latent growth function to meaningful cell-type–specific proliferation patterns.

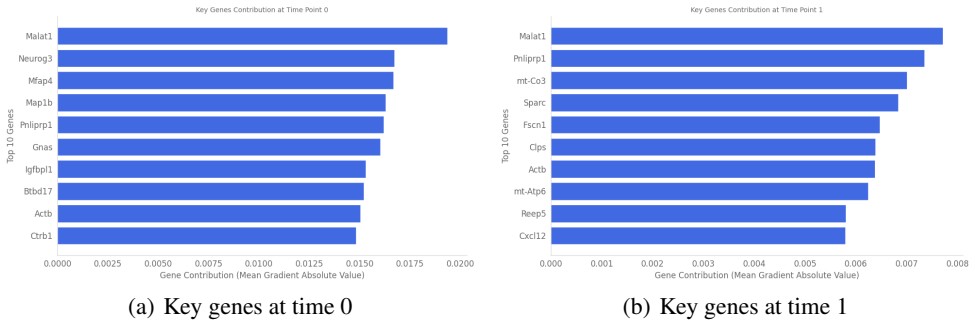

(a) Key genes at time 0                    (b) Key genes at time 1

Figure A-18: Key genes identified by $g_w(x, t)$ at time 0 (a) and 1 (b).

## C   Broader Impacts

Our method provides a scalable and efficient framework for modeling cellular dynamics, enabling trajectory reconstruction and growth rate inference in high-dimensional single-cell datasets. In biomedical and clinical settings, such capabilities can facilitate a deeper understanding of developmental processes, disease progression, and response to treatment at a single-cell resolution. For example, modeling differentiation trajectories and proliferation patterns of stem or immune cells could inform therapeutic strategies in cancer, regenerative medicine, and immunotherapy.

However, the proposed method is inherently data-driven and relies on statistical patterns learned from observational data. As such, it may produce biologically implausible trajectories or growth behaviors that conflict with known biological priors, especially when the training data is noisy, biased, or incomplete. This could potentially lead to misleading interpretations or incorrect clinical hypotheses if not carefully validated by domain experts. Therefore, we emphasize that any downstream medical conclusions drawn from the model's output should be interpreted with caution and in conjunction with biological prior knowledge and experimental validation.

