# OpenReview forum: "Joint Velocity-Growth Flow Matching for Single-Cell Dynamics Modeling"
_NeurIPS.cc/2025/Conference — NeurIPS 2025 poster_

### Official Review · Reviewer_G6tj · 2025-06-29

**Clarity:** 4
**Significance:** 4
**Originality:** 4
**Rating:** 6
**Confidence:** 5

**Summary:**

This paper proposes VGFM (Velocity-Growth Flow Matching), a novel method for modeling single-cell dynamics from snapshot data that accounts for both state transitions and mass variation (e.g., due to cell proliferation or death). The authors introduce a dynamic reinterpretation of semi-relaxed optimal transport (SROT) as a two-phase process: first, mass growth occurs (modeled via a growth function), and then cells transition in state space (modeled via a velocity field). From this decomposition, the authors derive closed-form supervision targets for learning velocity and growth via neural networks using a flow matching approach. An additional distribution fitting loss is incorporated to improve the match between predicted and observed snapshots. The method outperforms existing state-of-the-art (SOTA) methods on both synthetic and real datasets, with superior accuracy, interpretability, and scalability.

**Questions:**

- Could the authors provide correlation analyses between predicted growth rates and known proliferation markers or synthetic ground truth?
-  Have the authors considered estimating uncertainty in predicted velocity/growth (e.g., via dropout or ensembles)?
-  Can the method be extended to scenarios where timepoint alignment is unknown or uncertain, possibly by integrating OT-based time alignment?
- Effect of Sampling Bias: Since the method relies on observed cell counts for growth learning, have the authors tested robustness under non-uniform or subsampled cell distributions across time?

**Ethical Concerns:**

["NO or VERY MINOR ethics concerns only"]

**Final Justification:**

I appreciate the authors' effort to clarify their papers and address my concerns. As my concerns have been addressed, I would like to increase the rate by one score (from 5 to 6).

**Limitations:**

Yes
Additional suggestions:
- Consider including diagnostics or visualizations of growth error spatially or per-cell type.
- Investigate how growth modeling could be regularized or debiased using biological priors.

**Paper Formatting Concerns:**

All good.

**Quality:**

4

**Strengths And Weaknesses:**

Strengths:
- The paper is technically sound and builds on a solid mathematical foundation (SROT and continuity equations).
- The formulation of joint mass and state dynamics is well-posed, with provable equivalence to SROT objectives (Proposition 1, Theorem 1).
- Empirical results are strong across diverse settings, including high-dimensional synthetic data and real scRNA-seq datasets.
- Ablation studies and comparisons to unbalanced transport baselines (e.g., DeepRUOT, TIGON) are well done and validate all components.

Weeknesses:
- The estimation of the Monge map via barycentric projection introduces approximation error, especially when entropy regularization is non-negligible.
- The growth function $g(x, t)$ is derived from empirical densities and may absorb technical artifacts (sampling bias, dropout), but this is not fully analyzed.
- The method assumes aligned timepoint snapshots, limiting its direct application to asynchronous or pseudotime datasets.

---

> ### Author Rebuttal · Authors · 2025-07-31
>
> We thank the reviewer for the positive comments and insightful questions.
>
> ### **Q1: Could the authors provide correlation analyses between predicted growth rates and known proliferation markers or synthetic ground truth?**
>
> As suggested, we implement correlation analysis between the predicted growth rate and the ground truth to verify the accuracy of our growth rate modeling and learning as follows. We choose the Simulation Gene Data (Appendix B.2, Page 19) which is generated from the given dynamic systems (eq. (A-23), Page 19). The ground truth of growth rate can also be calculated analytically. Therefore, we can conduct **correlation analysis on the out-of-distribution (OOD) time points**, which are not included in the training data, to verify the generalization ability of VGFM. Results are reported in the following Table r4-1.
>
> Table r4-1: correlation analysis on four OOD time points on Simulation Gene Data
>
>
> | OOD time  | Pearson correlation coefficient |
> | :-------: | :-----------------------------: |
> | $t_{0.5}$ |              0.980              |
> | $t_{1.5}$ |              0.992              |
> | $t_{2.5}$ |              0.995              |
> | $t_{3.5}$ |              0.996              |
>
> The growth rate predicted by our VGFM shows a high correlation with the ground truth, and we will add this analysis in the Appendix.
>
> ### **Q2: Have the authors considered estimating uncertainty in predicted velocity/growth (e.g., via dropout or ensembles)?**
>
> While our current model is deterministic, dropout can be applied to our velocity/growth networks to obtain uncertainty.  During inference, we calculate the averaged variance of each dimension of velocity and growth over all the points across different branches (i.e., cell types) and times. We experiment on Simulation Gene Data (Appendix B.2), set dropout rate as 0.1 and execute 5 runs on the velocity/growth network during inference for all the points. Results are shown in the following Table r4-2.
>
> Table r4-2: uncertainty estimation of predicted velocity (2-dimensional) and growth on Simulation Gene Data. var_x1/var_x2  and var_g are the average variance of $1$-st/$2$-nd dimension of velocity and growth respectively.
>
> | branch | time | var_x1 (1e-3) | var_x2 (1e-3) | var_g (1e-3) |
> | ------ | ---- | ------------- | ------------- | ------------ |
> | 0      | 0    | 0.1080        | 0.1000        | 0.0550       |
> | 0      | 1    | 0.0770        | 0.0600        | 0.0660       |
> | 0      | 2    | 0.0660        | 0.0670        | 0.1100       |
> | 0      | 3    | 0.0850        | 0.0820        | 0.1910       |
> | 0      | 4    | 0.1220        | 0.1370        | 0.3500       |
> | 1      | 0    | 2.9320        | 6.5010        | 0.2510       |
> | 1      | 1    | 3.0230        | 4.2870        | 0.3060       |
> | 1      | 2    | 2.0410        | 1.9780        | 0.5590       |
> | 1      | 3    | 0.9530        | 0.9760        | 0.7660       |
> | 1      | 4    | 0.2800        | 0.3100        | 1.1680       |
>
>
> We analyze the results at both spatial and temporal levels.
>
> **Spatial level:** In the Simulation Gene Data, cells of branch 0 relatively stand still with little variation on state and counts. The result shows that the variance of $v_1$, $v_2$ and $g$ in this branch is small, indicating low uncertainty, while in branch 1 the variances is larger, indicating relatively higher uncertainty.
>
> **Temporal level:** In the Simulation Gene Data, cells undergo significant state transitions and quantity changes in the initial phase, and the magnitude of the change decreases thereafter. Accordingly, the variance of $v_1,v_2$ and $g$ also reflects this trend, confirming our uncertainty estimation.
>
> We will include these results as well as visulize the heatmap of uncertainty in the state space at each time points  in appendix.
>
>
>
>
> ### **Q3: Can the method be extended to scenarios where timepoint alignment is unknown or uncertain, possibly by integrating OT-based time alignment?**
>
> Our current method assumes known time labels. In the absence of such labels, one possible extension is to first infer pseudotime using standard approaches (e.g., Monocle, Slingshot), then apply OT-based alignment between discretized pseudotime segments to estimate cell transitions and population growth. Alternatively, traditional pseudotime methods typically rely on low-dimensional embeddings (e.g., PCA, UMAP) followed by trajectory reconstruction. Inspired by this, we are also interested in future work that jointly learns cell evolution trajectories, velocity fields, and growth dynamics directly in the latent space, integrating pseudotime inference and dynamic modeling into a unified OT-based framework.
>
> ### **Q4: Effect of Sampling Bias: Since the method relies on observed cell counts for growth learning, have the authors tested robustness under non-uniform or subsampled cell distributions across time?**
>
> In the cell dynamics reconstruction, we usually assume that the observed dataset accurately reflects the evolution of true cell distribution across time. So the observed cell counts is consistent with the fact of cell proliferation and death. It's also practically meaningful to evaluate the robustness of VGFM trained on the biased observed dataset.
>
> For the experiment settings, we use Simulation Gene Data (Appendix B.2) which has two spacial branches. To generate the non-uniform or subsampled data, we resample the original data with perturbed branch ratio. Specifically, we randomly select perturbation factor where $a_t\sim\mathcal{N}(0,\sigma^2)$ for time $t$. Then if the original  branch ratio is $s_t:1-s_t$, the resampled branch ratios become $s_t-|a_t|:1-s_t+|a_t|$ . $\sigma$ indicates the bias level. We train the model on the resampled data and evaluate on the original data, to testify the robustness of VGFM.
>
> Table r4-3: results of training on sampling bias on Simulation Gene Data with merics ($W_1$/RME) averaged over three random seeds.
>
> | bias factor $\sigma$ |     t1      |     t2      |     t3      |     t4      |
> | :------------------: | :---------: | :---------: | :---------: | :---------: |
> |     0 (unbiased)     | 0.041/0.007 | 0.053/0.005 | 0.038/0.007 | 0.040/0.008 |
> | 0.1  | 0.074/0.035 | 0.094/0.051 | 0.075/0.063 | 0.077/0.032 |
> | 0.2  | 0.085/0.059 | 0.113/0.073 | 0.134/0.062 | 0.123/0.125 |
>
> We also experiment on the real-world dataset EB (Appendix B.6). Since this dataset doesn't have obvious spatial branches, we take a different dropout method to construct the biased dataset and likewise **control the bias level with a factor $\eta$**. Specifically, we sample several $a_t\sim \mathcal{U}([0,\eta])$ with different bias factors, where $\mathcal{U}([0,\eta])$ is the uniform distribution on $[0,\eta]$. Then, the points at time $t$ with a proportion of $a_t$ are randomly removed from the dataset. In such way, we can get a biased version of EB dataset.
>
> Table r4-4: results of training on the biased EB datasets with merics ($W_1$/RME) averaged over three random seeds.
> | bias factor $\eta$ |     t1      |     t2      |     t3      |     t4      |
> | :----------------: | :---------: | :---------: | :---------: | :---------: |
> |    0 (unbiased)    | 7.951/0.039 | 8.747/0.042 | 9.244/0.019 | 9.620/0.044 |
> |        0.1         | 7.981/0.129 | 8.825/0.118 | 9.219/0.108 | 9.632/0.085 |
> |        0.2         | 8.019/0.072 | 8.908/0.165 | 9.330/0.111 | 9.723/0.181 |
>
> Tables r4-3 and r4-4 indicates that when the bias level is relatively low ($\sigma\le 0.1$ or $\eta\le 0.1$), our VGFM seems robust to the sampling bias. But if the bias level is high, the perfromance degrades. For such cases, we believe that with more biology prior beyond the cell counts integrated into growth modeling, our method could be more robust to the bias obervation. We will include these experimental results and the discussions in the appendix.
>
> ### **Q5: The estimation of the Monge map via barycentric projection introduces approximation error, especially when entropy regularization is non-negligible.**
>
> As stated in Appendix B.1 (Line 574-576, Page 18), we try to choose the entropy regularization coefficient $\varepsilon$ as small as possible to ensure it can approximate the exact optimal transport and monge map, while ensuring numerical stability and computational efficiency. Extensive experiments (Table 1, Table 2, Page 8) demonstrate the advantages of our VGFM despite the approximation error of exact optimal transport and Monge map.  We acknowledge that theoretical research of this approximation remains an open and intereting direction, and we thank the reviewer for highlighting it.
>
> ### **Q6: On limitations**
>
> For the first point, we will consider adding visualizations of growth error, both spatially and per cell type, in the appendix for the simulated gene dataset. However, for real datasets, growth rates are generally not directly observable, which limits such an analysis.
>
> For the second point, certain existing databases, such as UniProtKB, record genes that are potentially related to growth. We can leverage this information by comparing the learned $g$ with these known growth-associated genes, and incorporate such comparison as a regularization term during training to regularize or debias the growth modeling process.

---

> > ### Comment · Reviewer_G6tj · 2025-08-04
> >
> > I appreciate the authors' effort to clarify their papers and address my concerns. As my concerns have been addressed, I would like to increase the rate by one score (from 5 to 6).

---

> > > ### Author Response · Authors · 2025-08-04
> > >
> > > Thank you again for the valuable comments and we are very happy to address all the concerns. Your insightful and constructive comments have significantly enhanced our work, and we will certainly integrate your suggestions into the revised version of the paper.

---

### Official Review · Reviewer_DyDD · 2025-07-01

**Clarity:** 4
**Significance:** 4
**Originality:** 3
**Rating:** 5
**Confidence:** 4

**Summary:**

This paper proposes a flow and growth model for modeling interpolation of unbalanced snapshots of an evolving distribution. The evolution is derived from a two-stage process of growth and then flow, with the assumption that $p_0$ and the marginal of the coupling for the flow part have the same support. The resulting growth factor and flow vector field is then mixed to occur simultaneously, and proven to result in $p_1$.

Implementation of this framework is done via learning of the vector field and growth weight parameterized as neural networks. Training is done with sample access to the snapshot distributions and solving entropy-regularized semi-relaxed OT between the discrete distributions. An additional $W_1$ loss is also added to the result of the joint growth and flow model to improve alignment to the snapshots. Empirical performance is shown to outperform SOTA methods on synthetic and real datasets in achieving alignment to the snapshots.

**Questions:**

1. See point 1 in Weaknesses.
2. Is there a problem with non-differentiability of $W_1$ in the pipeline?
3. For Eq. 5, is there a reason you couldn't incorporate the energy from A.4 in order to get a unique specification for $g_t$? It would nice if it
mirrored the uniqueness of the velocity field from the Benamou-Brenier formulation.
4. Will an implementation be released?

**Ethical Concerns:**

["NO or VERY MINOR ethics concerns only"]

**Limitations:**

Yes, though Weakness point 1 should be addressed in some way.

**Quality:**

3

**Strengths And Weaknesses:**

Strengths:
1. Good theoretical justification of the approach is given.
2. The method is relatively simple, but nontrivially combines existing tools for entropically-regularized transport and neural ODEs.
3. Empirical performance and runtime seems to clearly beat existing methods.

Weaknesses:
1. It still seems unclear to me if the suggested model accurately reflects the potential behavior of genetic populations of cells. In particular, is it not possible for novel populations of cells to grow or appear outside the support of the current (or initial distribution) $p_0$? Or for whole populations to just disappear, and to have the remaining population be transported to the next snapshot? It would nice to comment on this or address this.
2. As noted by the authors, the method is not fully simulation-free due to the additional $W_1$ loss against snapshots.

---

> ### Author Rebuttal · Authors · 2025-07-31
>
> We sincerely thank the reviewer for the positive comments and insightful questions.
>
> ### **Q1: It still seems unclear to me if the suggested model accurately reflects the potential behavior of genetic populations of cells. In particular, is it not possible for novel populations of cells to grow or appear outside the support of the current (or initial distribution) ? Or for whole populations to just disappear, and to have the remaining population be transported to the next snapshot?**
>
> Cell differentiation naturally causes transitions in the state space, which can lead to changes in both the support set and the population size at the next time point. Our proposed VGFM aims to model such dynamics.
>
> It is a particularly challenging scenario that part of the population in a future snapshot does not originate from the first snapshot, but gradually emerges at some intermediate time from outside the support of the earlier distribution. From our current biological understanding, such a “spontaneous appearance” of an entirely novel population without any lineage connection is highly unlikely, as new cell populations generally arise through differentiation from existing ones. That said, if such a situation were to occur, we acknowledge that VGFM in its current form could not handle it, as it assumes that the population at the next snapshot evolves entirely from the previous one.
>
> For the second scenario mentioned, where a certain population disappears while the remaining population transitions to the next snapshot, VGFM can indeed identify such disappearing populations via the semi-relaxed OT plan, as these cells would have vanishing future weights. However, since such cells are not sampled during training, their growth rates and velocity fields could not be learned. Handling this case may require additional constraints, such as enforcing the velocity field to be zero and the growth rate to be very small in those regions, or incorporating biological priors to determine the likely fate and disappearance time of such populations.
>
>
>
>
> ### **Q2: Is there a problem with non-differentiability in the pipeline?**
>
> We appreciate the reviewer’s careful observation. We now clarify how the $W_1$ loss is computed. In practice, we use the `ot.emd()` function from the `pot` library to compute the transport plan $\pi$ in $W_1$. This function solves for $\pi$ using a network flow algorithm, which is not compatible with PyTorch’s automatic differentiation. However, the gradients of the cost function $c$ can still be backpropagated. This strategy is also adopted in the implementations of Zhang *et al.* (ICLR 2025) and Huget *et al.* (NeurIPS 2022). Therefore, the gradient of $W_1$ w.r.t. network parameters is partial rather than full. Nevertheless, this does not affect the effectiveness of our method nor the feasibility of computation, as demonstrated by our results (see sec. 4). Furthermore, in the rebuttal to reviewer kHRn (Please refer to Q3 of response to reviewer kHRn), we also explored replacing $W_1$ with the Sinkhorn divergence (Feydy *et al.*, AISTATS 2019), which is fully differentiable and enjoys many favorable properties. The Sinkhorn divergence and $W_1$ yield similar results. The corresponding results and discussion are provided in our response to Reviewer kHRn.
>
> ### **Q3: For Eq. 5, is there a reason you couldn't incorporate the energy from A.4 in order to get a unique specification for $g_t$?**
>
> The static semi-relaxed optimal transport in Eq. 4 is a **strictly convex** optimization problem: the first term is linear, the second term $\mathrm{KL}(\cdot \| q)$ is convex, and the constraints are linear. Therefore, the optimal transport plan $\pi^\ast$ is **unique**, which in turn makes $p_\lambda$ unique.  Given the additional constraint in A.4, we can then uniquely determine $g_t$. We aim develop an dynamic understanding in Eq. (5) of the static semi-relaxed OT in Eq. (4). Directly adding the $L^2$ energy term into the objective function of Eq. (5) would break its equivalence with Eq. (4). Instead, we treat it as a **constraint** to select $g_t$.
>
>
> ### **Q4: Will an implementation be released?**
>
> Yes. We are currently organizing and cleaning up the code, and it will be released publicly upon acceptance of the paper.
>
> ### **Q5: Introducing distribution fitting loss needs simulations.**
>
> Indeed, our method is not fully simulation-free due to the additional distribution fitting loss, we stress the importance of distribution fitting loss:
>
> **Necessity of distribution fitting loss:** Training the flow matching model only at the level of velocity and growth regression without an explicit global distribution-level fitting constraint may lead to inaccuracy distribution learning. Specifically, since the theoretically optimal velocity $v\_\theta(x,t)=\mathbb{E}\_{z|(x,t)}v_\theta(x,t|z)$ in flow matching, there may exist a possible velocity average due to the expectation, and the expected velocity may deviate from the real velocity. Introducing the distribution fitting loss will enforce the global distributional match constraint over the evolved population of cells.
>
>
> **Computational Efficiency Comparison:** As the ablation study (Table 3 in Sect. 4.1, Page 9) shows, the computation cost needed for VGFM (13 mins) is **much less** than the totally simulation-based method DeepRUOT (90 mins), although VGFM sacrifices some computation efficiency compared with the totally simulation-free VGFM (w/o $\mathcal{L}\_{\rm OT}$). At the same time, VGFM achieves the **best results** on the dynamics reconstruction among the totally simulation-based and totally simulation-free methods.

---

### Official Review · Reviewer_fzn6 · 2025-07-05

**Clarity:** 3
**Significance:** 2
**Originality:** 3
**Rating:** 5
**Confidence:** 3

**Summary:**

The paper proposes VGFM (Joint Velocity-Growth Flow Matching), a scalable, simulation-free approach to modeling single-cell dynamics from unpaired, unbalanced snapshot data. By framing the dynamics as a two-period semi-relaxed optimal transport problem, the authors decouple and then recombine velocity and growth fields into a joint dynamic system. The method uses neural networks to parameterize these fields and optimizes a flow-matching loss with an additional distribution-fitting term. Experiments demonstrate VGFM’s performance on synthetic Gaussian data, a 5D EB dataset, and CITE-seq data.

**Questions:**

- The scalability of VGFM is very interesting. Would training directly on gene-level counts (say 1k-2k highly variable genes) be feasible? Do the authors have an intuition how the sparsity of scRNA-seq would be handled by VGFM?

- Could you elaborate on the motivation behind the distribution-fitting loss? I’m curious whether it was derived from a principled rationale or chosen empirically.

- I would be interested to hear the authors’ perspective on the biological significance of explicitly modeling growth effects. Methodologically, this choice is well motivated and aligns well with the nature of the data.

- Could the authors elaborate on how their model could be made conditional on covariates (e.g., cell type, batch)? Incorporating conditioning is important in single-cell modeling and would enable meaningful applications such as data augmentation and batch correction.

&nbsp;
_____
I appreciate the authors’ effort in addressing an important research question and look forward to their clarifications and responses during the rebuttal phase. I am happy to raise my score if the authors adequately address my questions and concerns.

**Ethical Concerns:**

["NO or VERY MINOR ethics concerns only"]

**Final Justification:**

The authors addressed most of my concerns and put great effort into the rebuttal. I believe that the additional experiments really benefit the paper and demonstrate the potential of the VGFM method.

**Limitations:**

While the authors acknowledge some limitations of their approach, the discussion remains focused on technical aspects. It would strengthen the manuscript to also address broader challenges in single-cell modeling (noise structure, sparsity, overdispersion, data acquisition). A discussion of where VGFM is likely to succeed and where it may face limitations in the context of real single-cell data would provide valuable perspective (for example, the scalability claim is only demonstrated on synthetic Gaussian data rather than realistic gene-level scRNA-seq datasets).

**Paper Formatting Concerns:**

Minor issues: typo in heading L126; Table 2 does not define the evaluation metric clearly; L233–234 lacks dataset details as pointed out above.

**Quality:**

3

**Strengths And Weaknesses:**

### Strengths
- [Originality] The formulation of the dynamics via semi-relaxed OT with separate velocity and growth fields is novel and well-grounded. The manuscript clearly explains the derivation of VGFM and its training objectives.
- [Significance] The proposed method allows training on very high-dimensional synthetic data (up to 1000D) efficiently, outperforming prior work in terms of scalability.

### Weaknesses
- [Significance] My biggest concern is the misalignment of motivation and experiments. Although the method is explicitly motivated as being tailored for single-cell dynamics, the experiments largely fail to demonstrate its merits in this domain. The synthetic Gaussian and low-dimensional EB experiments are insufficient to validate biological relevance or single-cell-specific challenges. For this, it is essential to analyse the modeled samples (i.e. gene) space. I would suggest to investigate VGFM in the following contexts:
     - A comparison of the gene-wise empirical mean-variance trend with that of real data and an analysis of the modeled sparsity in gene space, cf. 5.1 in [1]
     - The inclusion of the MEF dataset [2] as well as the analysis of the decoded space as done in 5.2 of [3]
     - An analysis on the Pancreas dataset [4], checking that with VGFM one can identify all terminal states as well as investigating the dynamics of linage drivers. For the linage drivers, in particular, it would be interesting if the growth modelling enables more accurate modeling of the real data, cf. [3].
- [Quality] I encourage the authors are encouraged to discuss key characteristics of scRNA-seq data, such as sparsity, overdispersion, and its negative binomial noise structure. Since VGFM can readily be combined with alternatives to PCA that better reflect the data’s nature, I encourage the authors to consider and elaborate on this connection (incorporating an NB noise model may also be relevant for the additional experiments suggested above).
- [Clarity] Key dataset details (e.g., number of cells, timepoints, gene dimensions; see L233–234) are missing. Including this information would take little space but greatly improve the clarity of the experimental setup for the reader.
- [Minor] The additional distribution loss term (L68–69) appears a bit ad-hoc and lacks justification (beyond empirical performance) compared to the VGFM part.

&nbsp;

_____

- [1] Palma et al. "Multi-Modal and Multi-Attribute Generation of Single Cells with CFGen", ICLR 2025
- [2] Schiebinger et al. "Optimal-transport analysis of single-cell gene expression identifies developmental trajectories in reprogramming", Cell, 176(4):928–943, 2019.
- [3] Palma et al. "Enforcing Latent Euclidean Geometry in Single-Cell VAEs for Manifold Interpolation", ICML 2025.
- [4] Bastidas-Ponce et al. "Comprehensive single cell mRNA profiling reveals a detailed roadmap for pancreatic endocrinogenesis" Development, 146(12):dev173849, 2019.

---

> ### Author Rebuttal · Authors · 2025-07-31
>
> We sincerely appreciate your valuable and insightful suggestions. We have conducted additional experiments and will **include results and discussions in our manuscript and appendix**.
>
> ### **Q1: The scalability of VGFM on real-world high-dimensional gene-space datasets.**
>
> To further explore the scalability of VGFM, we applied our model and compared methods explicitly modeling $g\_t(x)$ (Sha *et al*., 2024; Zhang *et al*., 2025) to the Pancreas dataset with 1000 highly variable genes.
>
> + **Stable training loss curve.** We observed that VGFM is the only method showing a steadily decreasing training loss, both for $\mathcal{L}\_{\mathrm{VGFM}}$ and $\mathcal{L}\_{\mathrm{OT}}$. We compute $W\_1$ at day 15.5 between real data and generated data from VGFM and its variant VGFM (w/o) $\mathcal{L}\_{\mathrm{OT}}$, as shown in the table below:
>
> Table r2-1: $W_1$ at day 15.5 between real data and generated data from VGFM and its variant.
>
> |Method|$W_1$|
> |-|-|
> |VGFM (w/o $\mathcal{L}_{\mathrm{OT}})$|24.583|
> |VGFM|21.026|
>
> This indicates that both loss terms are still effective in high-dimensional real-world data.
>
> + **Analysis on mean and variance.** We calculated the means and variances of the real and generated data at day 15.5 and plotted the corresponding trend. The results show that the generated samples closely follow the mean–variance trend of the real data. To present our results, we provide a table showing binned statistics, summarizing the distribution of real and generated data across different mean and variance intervals.
>
>
> Table r2-2: Distribution of real and generated data across different mean and variance intervals.
> |Mean intervals|Generated data count|Real data count|Variance intervals|Generated data count|Real data count|
> |-|-|-|-|-|-|
> |[0.03,0.59)|512|514|[0.02,0.52)|829|765|
> |[0.59,1.15)|278|269|[0.52,1.01)|119|164|
> |[1.15,1.71)|87|88|[1.01,1.50)|25|32|
> |[1.71,2.28)|48|48|[1.50,1.99)|15|17|
> |[2.28,2.84)|46|50|[1.99,2.48)|5|9|
> |[2.84,3.40)|21|21|[2.48,2.97)|5|6|
> |[3.40,3.96)|2|4|[2.97,3.46)|1|2|
> |[3.96,4.53)|1|0|[3.46,3.96)|0|3|
> |[4.53,5.09)|4|5|[3.96,4.45)|0|1|
> |[5.09,5.65)|1|1|[4.45,4.94)|1|1|
>
> As shown in the table, the distribution of means in the generated data exhibits a high degree of consistency with that of the real data, while the variances are generally slightly smaller than those of the real data but still remain reasonably accurate.
>
> + **Analysis of the sparsity.**  Our current paper is applied to PCA-projected single-cell data, with the main contributions lying in the theoretical formulation and algorithmic design for the **joint dynamic modeling of $g_t$ and $v_t$**, from which we derive our flow-matching-based approach. In its present form, VGFM does not explicitly impose sparsity constraints on the data, generating data not as sparse as the real data. Nevertheless, recent studies have introduced encoder–decoder frameworks that operate in a latent space while explicitly modeling single-cell gene expression properties such as sparsity and the negative binomial distribution (Palma *et al.*, 2025). By integrating VGFM into such a framework, for example, using advanced encoders to map raw count data into a latent space and appropriately designing decoders to reconstruct gene counts, one could learn the growth function from OT plans in the latent space with more accurate metrics beyond the Euclidean cost, while simultaneously preserving biologically relevant statistical properties. We believe this **direct extension to raw data with explicit sparsity and negative binomial modeling** is a promising direction, deserving thorough investigation in future work. We will cite these suggested papers and discuss them in the manuscript.
>
> + **Computing fate probabilities.** We follow Eyring *et al*. (2024) to compute the averaged fate probabilities of each ground truth transition. Notably, in almost each transition, VGFM ranked among the top three performers, despite operating directly in the 1,000-dimensional gene expression space, whereas all other methods were evaluated in a 10–50-dimensional PCA space, as done in Eyring *et al*. (2024). This might indicate that VGFM can achieve biological interpretability even in high-dimensional settings.
>
> Table r2-3:Averaged fate probabilities of each ground truth transition.
> |Model|dim|FA→A|A→A|FB→B|B→B|FD→D|D→D|FE→E|E→E|NL→ED|DU→DU|T→AC|AC→AC|
> |-|-|-|-|-|-|-|-|-|-|-|-|-|-|
> |TrajectoryNet|10 PCs|0.07|0.46|0.07|0.39|0.11|0.75(3)|0.10|0.52|0.01|0.32|0.79(3)|0.99(2)|
> |scVelo|50 PCs|0.79(1)|0.80(1)|0.30(1)|0.61|0.03|0.52|0.04|0.56(3)|1.00(1)|0.32|0.04|0.90|
> |WOT|50 PCs|0.37|0.62(2)|0.18(2)|0.44|0.19(3)|0.74|0.55(1)|0.49|0.50|0.82(2)|0.48|0.84|
> |OT-ICNN|50 PCs|0.51(2)|0.54|0.11|0.70(3)|0.39(2)|0.94(2)|0.48|0.62(2)|0.08|1.00(1)|1.00(1)|1.00(1)|
> |UOT-ICNN|50 PCs|0.25|0.60|0.29(2)|0.75(1)|0.78(1)|0.99(1)|0.36|0.55|0.73(2)|0.59|1.00(1)|1.00(1)|
> |VGFM|1000 HVGs|0.42(3)|0.80(1)|0.13|0.63(3)|0.14|0.70|0.64(1)|0.80(1)|0.73(2)|0.66(3)|0.39|0.91|
>
> + **Interpretable learned growth function.** We highlight that our main contribution lies in modeling and training the growth function $g_t(x)$, with VGFM in the flow matching framework, to leverage snapshot mass change for learning $g_w(x,t)$. During training, the growth loss rapidly converges to a negligible value, much faster than the velocity loss. To further investigate $g_w(x,t)$, we follow Sha *et al.* (2024), for each time $t$, compute the element-wise absolute value $\left|\frac{\partial g(x,t)}{\partial x_i}\right|$ in the gene space for every cell, and average across cells to quantify each gene’s contribution to growth dynamics. From our analysis of $g_w(x,t)$, the three genes with the largest absolute gradients are **Pnliprp1, Clps, and Ctrb1**. These genes are well-known markers of Acinar cells and are highly expressed in the exocrine pancreas. Notably, at this time point, cells from the non-endocrine branch, particularly Acinar cells, are much more abundant at later stages due to high proliferation rates. This alignment between the learned genes and known biological processes indicates that VGFM captures growth dynamics in a biologically interpretable manner, successfully linking the learned growth function to meaningful cell-type-specific proliferation patterns.
>
> ### **Q2: The motivation behind the distribution-fitting loss.**
>
> Our ultimate goal is to generate data close to the real data distribution. Although flow matching is a powerful generative modeling framework, its learned velocity field is the expectation of the conditional velocity field, i.e.,$v\_\theta(x,t)=\mathbb{E}\_{z|(x,t)}v_\theta(x,t|z)$， where $z=(x_0,x_1)$ is the entropic semi-relaxed ot plan in our setting, which is a **soft coupling**. For the same $x_0$, $x_1$ may come from different branches of the trajectory. This bias could accumulate during generation, possibly leading to deviation from the target distribution. To address this bias, we introduce a distribution-fitting loss that directly measures how well the generated samples match the empirical data distribution at each time point. As done in the ablation study (Page 9, Table 3) on loss terms in our paper, this complements the flow-matching objective and ensures that even if the velocity field is biased, the model still adjusts its dynamics to match the correct data distribution.
>
> ### **Q3: The biological significance of explicitly modeling growth effects.**
> We appreciate the reviewer’s positive feedback on our methodological choice of explicitly modeling growth effects. From a biological perspective, cell development is inherently a coupled process of **state transitions** and **mass changes**. Ignoring the growth component may lead to erroneous state transition inference, for example, reconstructing cross-branch trajectories that are biologically implausible, as we demonstrated in the simulation gene dataset experiment in our paper.  Moreover, modeling growth explicitly enables downstream biological interpretation. By computing element-wise absolute value $\left|\frac{\partial g(x,t)}{\partial x_i}\right|$ in the gene space, one can **identify growth-related genes and proliferative cell populations**, as done in Sha *et al.*,2024. However, their approach relies on autoencoder-based dimensionality reduction.
> In our response to Q1, we have analyzed our learned $g_w(x,t)$, which is trained directly on gene space, on the Pancreas dataset. We successfully identified key genes such as Pnliprp1, Clps, and Ctrb1, which are known to be associated with proliferating Acinar cells. This demonstrates that VGFM also yields biologically meaningful insights into cell population dynamics, even in high-dimensional settings where previous methods face limitations.
>
> ### **Q4: How VGFM could be made conditional on covariates.**
> A natural approach is to encode cell type as a one-hot vector $c$ and integrate it into both $v_\theta(x,t,c)$ and $g_w(x,t,c)$. The main challenge lies in determining the coupling between cells of the same type across two time points. This becomes considerably easier if prior knowledge of the transition fate for each cell type is available. While VGFM can identify proliferating cell types (as demonstrated in the Pancreas dataset), incorporating such prior knowledge could improve model accuracy. Moreover, VGFM could be extended to model perturbation responses conditioned on different drugs or gene perturbations. These embeddings could be generalized to unseen perturbations via the mode-of-action framework (Bunne *et al.*, 2022), enabling VGFM to predict which perturbations might facilitate or inhibit cell growth.
>
> ### **Q5: Key dataset details.**
>
> Thanks for your suggestions, we will add this table to our appendix.
>
> Table r2-4: Key dataset details.
>
> |Dataset|Timepoints|Cell counts|Dimensions|
> |-|-|-|-|
> Simulation Gene|5|3031|2|
> Dyngen|5|728|5|
> Gaussian|2|1900|1000
> Mouse Hematopoiesis|3|10998|2
> EB|5|16819|5,50|
> CITE|4|31240|5,50|
> Pancreas|2|18941|1000|

---

> > ### Comment · Area_Chair_UVRG · 2025-08-04
> > **Discussion reminder**
> >
> > Dear Reviewer,
> >
> > We are getting close to the end of the discussion phase. The authors have provided a detailed response to the reviews. When you have a moment, could you please take a look and post a brief comment?
> >
> > Specifically, it would be helpful to know if their response addresses your main concerns and whether it impacts your overall assessment. This feedback is invaluable for the discussion phase.
> >
> > Thank you for your time and effort on this.

---

> > ### Comment · Reviewer_fzn6 · 2025-08-05
> >
> > I thank the authors for the answers and efforts towards providing additional experiments. While the authors were able to answer most of my questions, some concerns regarding the experiments remain:
> > - I appreciate that the methods seems to works well in the setting of 1000HVGs, this is really promising. Have the authors checked what happens when fitted on 2k genes? Also, how does VGFM compare when fitted just on PCA space.
> > - Referring to Q3, the authors are correctly mentioning that biological insight is related to correctly fitting the gene space. I am wondering if the same genes can be identified when fitting VGFM on PCA space and then mapping back to gene space.
> > - Furthermore, I would strongly suggest to include the MEF dataset for trajectory reconstruction. The provided $W_1$ metric for the pancreas dataset is not readily comparable to the analysis as performed in 5.2 of [3] (please refer to my original review for details).

---

> > > ### Author Response · Authors · 2025-08-06
> > >
> > > We first address the first two questions, and we are actively working on the third one, to which we will respond before the deadline.
> > >
> > > ### Q1: Model performance on 2k gene space and 100 PCA space.
> > >
> > > As suggested, we further applied our model to the 2k-gene space of the Pancreas dataset.The results are reported in Tables r2-5. We found that the training remained still stable. Specifically, the PCA setting was even more stable than the other experiments.
> > >
> > > Table r2-5: $W_1$ at day 15.5 between real data and generated data from VGFM and its variant on 2k gene space and 100 PCA space.
> > > |Method|$W_1$ (2k gene space)|$W_1$ (100 PCA space)|
> > > |-|-|-|
> > > |VGFM (w/o $\mathcal{L}_{\mathrm{OT}})$|28.405|12.503|
> > > |VGFM|25.451|9.452|
> > >
> > >
> > > Table r2-6: Distribution of real and generated data across different mean and variance intervals on 2k gene space.
> > > | Mean intervals | Generated data count | Real data count | Variance intervals | Generated data count | Real data count |
> > > |-|-|-----------------|--------------------|----------------------|-----------------|
> > > | [0.00, 0.57)   | 1492                 | 1495            | [0.00, 0.49)       | 1805                 | 1748            |
> > > | [0.57, 1.13)   | 286                  | 284             | [0.49, 0.99)       | 136                  | 180             |
> > > | [1.13, 1.70)   | 95                   | 90              | [0.99, 1.48)       | 28                   | 32              |
> > > | [1.70, 2.26)   | 48                   | 49              | [1.48, 1.98)       | 17                   | 17              |
> > > | [2.26, 2.83)   | 48                   | 50              | [1.98, 2.47)       | 6                    | 9               |
> > > | [2.83, 3.39)   | 22                   | 21              | [2.47, 2.96)       | 3                    | 6               |
> > > | [3.39, 3.96)   | 3                    | 5               | [2.96, 3.46)       | 1                    | 3               |
> > > | [3.96, 4.52)   | 0                    | 0               | [3.46, 3.95)       | 3                    | 3               |
> > > | [4.52, 5.09)   | 5                    | 5               | [3.95, 4.44)       | 0                    | 1               |
> > > | [5.09, 5.65)   | 1                    | 1               | [4.44, 4.94)       | 1                    | 1               |
> > >
> > >
> > > For the *2k-gene space* setting, we report the count of real and generated data across different mean and variance intervals on 2k gene space in Table r2-6. We can see that the distribution of means in the generated data still exhibits a high degree of consistency with that of the real data, while the variances are generally slightly smaller than those of the real data but still remain reasonably accurate.  For the *PCA setting*, we can visualize the distributions of the generated and real data for each principal component, which will be presented in the appendix. We observe that our model incorporating the distribution fitting loss closely matches the distribution of each principal component.
> > >
> > > ### Q2：Identify genes in PCA space.
> > > For the 2k-gene setting, we adopt the same strategy described earlier to identify growth-related genes. The top-3 genes are the same as those of 1k setting, i.e., *Clps*, *Pnliprp1*, and *Ctrb1*, albeit in a different order, which demonstrates both the robustness of our model and its feasibility in a higher-dimensional gene space.
> > >
> > > For the PCA setting, since $X_{\mathrm{pca}} = X_{\mathrm{gene}} W^\top$, where $W$ is a projection matrix mapping from the PCA space of dimension $n$ to the gene space of dimension $m$, we have
> > > $\frac{\partial g(x,t)}{\partial x_{\mathrm{gene}}}
> > > = \frac{\partial g(x,t)}{\partial x_{\mathrm{pca}}} \frac{\partial x_{\mathrm{pca}}}{\partial x_{\mathrm{gene}}}
> > > = \frac{\partial g(x,t)}{\partial x_{\mathrm{pca}}} \frac{\partial (X_{\mathrm{gene}} W^\top)}{\partial x_{\mathrm{gene}}}
> > > = \frac{\partial g(x,t)}{\partial x_{\mathrm{pca}}} W.$
> > > Thus, it suffices to store the projection matrix $W$ during PCA to recover gene-level gradients.
> > >
> > > From the PCA-recovered gene space, we identify *Cck*, *Tmsb4x*, and *Spp1*, which do not exhibit strong biological interpretability. We conjecture that, as noted earlier, PCA may fail to capture the intrinsic properties of biological data, and its dimensionality reduction inevitably incurs information loss. In contrast, modeling directly in the gene space is more conducive to identifying biologically meaningful growth-related genes. This further underlines the importance and necessity of modeling in the gene space. Thanks to the experiments suggested by the reviewer, our VGFM seems to be the earliest success for performing key gene analysis directly in the gene space using the learned $g$, to the best of our knowledge.

---

> > > > ### Comment · Reviewer_fzn6 · 2025-08-06
> > > >
> > > > Thanks for providing these additional results. It is nice to see that VGFM seems to scale to large gene spaces. I suggest that the authors revise the presented experiments on scRNA-seq data based on the insights gained during the rebuttal and discussion period. Moreover:
> > > >
> > > > - When I referred to a PCA comparison I implicitly meant to extent r2-3.
> > > > - Still, I would like to see how VGFM performs on the MEF datasets (and ideally the EB dataset) in the mentioned evaluation setting of trajectory inference.
> > > >
> > > > Nevertheless, I will increase my score based on the additional results and experiments that were provided by the authors.

---

> > > > > ### Author Response · Authors · 2025-08-06
> > > > >
> > > > > We sincerely thank you for your insightful comments and valuable suggestions. We will continue our experiments on the MEF dataset and provide a detailed response before the deadline.

---

> > > > > ### Author Response · Authors · 2025-08-09
> > > > >
> > > > > ### Experiment on MEF dataset.
> > > > >
> > > > > We are so sorry for lacking the experiment on the MEF dataset in the previous response, due to the limited time for investigating this dataset. As suggested, we applied our model to the MEF dataset with 1479 highly variable genes, which contains over 160k cells and 39 time points.
> > > > >
> > > > > We noticed that the decoded space in [3] consists of discrete gene count data. We first located the GSE122662 dataset on the NIH website. However, the raw data there was highly fragmented without processing and integration. Due to limited time, we downloaded the preprocessed version provided in the Waddington OT [2] official website. In this version, the gene expression values are no longer in the discrete count space but have already undergone preprocessing. Due to time constraints, we performed our hold-out experiments on this preprocessed dataset and compared the results against OT-CFM (Tong et al.,2024), which also operates in the continuous space. As shown in Table r2-7, our model consistently achieves lower W1 distances at all hold-out time points compared to OT-CFM. This validates the effectiveness of our experimental approach.
> > > > >
> > > > >
> > > > >
> > > > > Table r2-7: W1 at hold-out timepoints of OT-CFM and VGFM. We denote the original 39 time points as $t=0$ to $t=38$.
> > > > > |models|$t=2$|$t=4$|$t=6$|$t=10$|$t=15$|$t=20$|$t=22$|$t=24$|$t=28$|$t=30$|$t=32$|$t=35$|$t=37$|
> > > > > |-|-|-|-|-|-|-|-|-|-|-|-|-|-|
> > > > > |OT-CFM|39.561|39.195|39.009|32.900|35.268|41.505|40.486|41,299|36.330|40.053|39.027|39.501|42.256
> > > > > |VGFM|37.514|38.862|38.175|32.430|34.717|40.560|39.891|39.840|35.732|39.448|38.987|39.112|40.336|
> > > > >
> > > > > In addition, we applied the learned growth function $g$ to highlight genes potentially associated with cell growth at different stages.  At around time point 10, we identified Uba52, Eif2s2, Ckb, and Nme2, which have been reported in contexts such as protein synthesis and cellular metabolism.  Around time point 20, we identified Aldoa, Gchfr, and Txnip, previously linked to metabolic and redox processes.  At time point 30, we identified Sparc and Ctsj, which have been mentioned in studies of extracellular matrix and protein processing.  The identification of such stage-specific genes, many of which have prior biological annotations, supports the interpretability of our model in capturing reprogramming dynamics in the MEF dataset.
> > > > >
> > > > >
> > > > >
> > > > > Our main contribution lies in jointly modeling the velocity field and growth function using flow matching, enabling direct analysis on gene space. In future work, we will continue to investigate the potential extensions of VGFM to discrete gene count data, combining the joint modeling of the velocity field and growth function with existing methods (e.g. latent representations in Palma et al.,2025) to facilitate accurate modeling in discrete gene count data, considering the negative binomial nature of gene count distributions will be an interesting and promising direction. We sincerely thank you for providing us with highly professional and constructive feedback during the rebuttal stage, which has offered us valuable perspectives for improving our work. We will include these results in the appendix and incorporate a thorough analysis of this aspect in the discussion section of the manuscript and cite the relevant literature accordingly.

---

> > > > > > ### Comment · Reviewer_fzn6 · 2025-08-09
> > > > > >
> > > > > > I appreciate the authors‘ efforts to include the experiment on the MEF dataset. While the comparison to the models I mention in my earlier responses is still not possible, I value the analysis provided give. The limited time of the rebuttal. I believe that the discussion pointed the authors in a valuable direction to demonstrate the capabilities and value of their method for the single-cell domain. For a revised version of the manuscript, a more extensive analysis of VGFM on the MEF dataset is desirable.
> > > > > >
> > > > > > For the moment, I do not have further concerns and I will (as already indicated) increase my score.  I recommend acceptance for this paper.

---

### Official Review · Reviewer_kHRn · 2025-07-12

**Clarity:** 3
**Significance:** 2
**Originality:** 3
**Rating:** 3
**Confidence:** 4

**Summary:**

The paper presents a new approach, called Velocity-Growth Flow Matching (VGFM), for learning single-cell population dynamics from snapshot data. Unlike existing methods that either rely on simulation-heavy training or ignore the mass imbalance due to cell proliferation and death, VGFM jointly models cell-state transitions (velocity) and mass variations (growth). The method leverages a theoretical foundation based on a "two-period" interpretation of semi-relaxed optimal transport, separating the processes of mass growth and cell-state transitions to derive an ideal dynamics formulation. Neural velocity and mass-variation functions are then trained to approximate these dynamics via flow matching, augmented with a Wasserstein-based distribution fitting loss. Extensive experiments on synthetic datasets (e.g., Gaussian mixtures and Dyngen) and real-world single-cell datasets demonstrate that VGFM reconstructs trajectories and growth dynamics, achieving superior performance compared to multiple strong baselines.

**Questions:**

- Could the authors provide a discussion regarding the influence of batch size on their method? As both the growth matching loss and the OT fitting loss rely explicitly on mini-batch optimal transport computations, this aspect is crucial. Statistically, we should expect that increasing the batch size would improve performance, as larger batches yield more accurate OT approximations and stable gradient estimates. Providing empirical evidence or a sensitivity analysis on how batch size affects model performance and stability would significantly strengthen the evaluation.

- Could the authors further elaborate on their motivation for choosing a time-independent growth function? Currently, the manuscript briefly states (lines 177–178): "For simplicity and ease of implementation, we choose a time-independent form...". However, it would be beneficial to explicitly discuss why this simplification—assuming that mass variations depend solely on spatial variables—is biologically or computationally justified. Clarifying whether this choice could potentially limit model expressiveness or generalization would improve the manuscript’s transparency and reader understanding.

- Could the authors clarify whether they rely on the Sinkhorn algorithm to compute the OT fitting loss (involving the Wasserstein-1 distance)? If so, do you choose a small entropic regularization parameter to closely approximate the true Wasserstein distance? Additionally, have you explored or considered using the Sinkhorn divergence (as introduced by  [Feydy et al., 2019]) instead of the standard entropic Wasserstein distance, given its favorable metric properties?

**Ethical Concerns:**

["NO or VERY MINOR ethics concerns only"]

**Final Justification:**

I appreciate the thoughtful and detailed responses provided by the authors. However, I remain not entirely convinced by the current methodological approach. I believe the paper would significantly benefit from additional revisions and resubmission in the future.

**Limitations:**

The authors clearly acknowledge important limitations of their work (section 5, page 9).

**Paper Formatting Concerns:**

NA.

**Quality:**

3

**Strengths And Weaknesses:**

**Strengths.**

This paper has several substantial strengths. First, it addresses an important problem—learning single-cell dynamics from unpaired snapshot data—while explicitly handling the biologically relevant challenge of mass imbalance. The proposed theoretical framing via semi-relaxed optimal transport is both elegant and well-motivated. Experimentally, the authors conduct thorough evaluations using synthetic and real-world datasets, demonstrating notable performance improvements over existing methods, especially in high-dimensional settings (up to 1,000 dimensions), where competing approaches significantly struggle. Additionally, the empirical validation provides compelling visualizations of the learned trajectories and growth rates.

**Weaknesses.**

- As acknowledged by the authors (lines 37–41), incorporating the OT-based distribution fitting loss renders the training procedure simulation-based. Specifically, this requires both simulating the underlying ODE dynamics and backpropagating through these simulations, making the method computationally cumbersome. This design choice diverges from recent advances in generative modeling, which explicitly aim for simulation-free training strategies to improve scalability and efficiency.

- Moreover, the authors do not explicitly describe how they compute the distribution fitting loss. Presumably, they rely on the Sinkhorn algorithm with Euclidean cost computed over mini-batches. If this is indeed the case, it should be clearly stated. This approach presents scalability concerns, as mini-batch OT approximations suffer substantially from the curse of dimensionality. A practical suggestion is to replace the Wasserstein-1 distance with a Sinkhorn divergence [Feydy et al., 2019], which possesses more favorable statistical properties. Nonetheless, even this improvement may not fully resolve scalability issues, particularly when dimensionality exceeds $d > 1,00$.

- The authors do not discuss the critical issue of batch size selection, despite the fact that two of the three loss components (growth function matching and OT distribution fitting) depend explicitly on mini-batch OT computations. Statistically, performance should improve as batch size increases. The current choice of a batch size of 256 for most experiments seems relatively small. Exploring batch sizes of 1,024–2,048 is recommended, especially given that Sinkhorn-type algorithms (including their unbalanced variants) remain computationally feasible at these scales. Furthermore, the authors propose partitioning computations into subsets (Appendix B.6) for larger datasets, which could negatively impact the quality and stability of the learned velocity and growth functions.

- Additionally, concerns arise regarding the stability and scalability of the proposed growth matching loss. While the theoretical justification for its introduction is appealing, the practical implementation—which involves regressing against the right marginal of empirical unbalanced OT couplings computed via mini-batches—is potentially unstable, particularly for smaller batch sizes. Specifically, estimating the growth function effectively involves approximating potentials from the dual unbalanced OT problem [Séjourné et al., 2024], a procedure known to suffer considerably from the curse of dimensionality. Moreover, this instability is exacerbated by the potential inconsistency in how mini-batches handle outliers; some points might have their mass reduced or discarded in one mini-batch, yet retained or increased in another.

- Finally, the notation throughout the paper, particularly in Sections 3.3 and 3.4, is often cumbersome and difficult to follow. Simplifying notation in these sections would significantly improve readability and accessibility.

---

> ### Author Rebuttal · Authors · 2025-07-31
>
> We sincerely appreciate your valuable and insightful suggestions.
> ### **Q1: Choice of distribution fitting loss diverges from simulation-free training.**
>
> **Necessity of distribution fitting loss:** Training the flow matching model only at the level of velocity and growth regression without an explicit global distribution-level fitting constraint may lead to inaccuracy distribution learning. Specifically, since the theoretically optimal velocity $v\_\theta(x,t)=\mathbb{E}\_{z|(x,t)}v_\theta(x,t|z)$ in flow matching, there may exist a possible velocity average due to the expectation, and the expected velocity may deviate from the real velocity. Introducing the distribution fitting loss will enforce the global distributional match constraint over the evolved population of cells.
>
>
> **Computational Efficiency Comparison:** As the ablation study (Table 3 in Sect. 4.1, Page 9) shows, the computation cost needed for VGFM (13 mins) is **much less** than the totally simulation-based method DeepRUOT (90 mins), although VGFM sacrifices some computation efficiency compared with the totally simulation-free VGFM (w/o $\mathcal{L}\_{\rm OT}$). At the same time, VGFM achieves the **best results** on the dynamics reconstruction among the totally simulation-based and totally simulation-free methods.
>
> ### **Q2: Influence of batch size.**
>
> We clarify that **our VGFM approach did NOT use mini-batch OT** either in the computation of the OT plan or the distribution fitting loss. Instead, we propose **an offline sampling strategy for OT plan computation** and its improved version tailored for large-scale datasets.
>
> **Offline sampling strategy**: **The OT plan between the adjacent snapshots for the whole population is precomputed and cached before training**, as **Algorithm 1 (Page 7)** shows. Then, in training of the flow matching, we sample a batch of paired timestep samples using this precomputed OT plan, alleviating the sampling bias. This strategy needs the computation of OT for the whole dataset. We further propose the **Big Batch strategy** in Page 22, Line 668. Specifically, we  **uniformly partition** the data at each time point into $n$ subsets as Big Batches. Then we precompute and cache the $n$ OT plans for these Big Batches and randomly select one Big Batch for constructing batches in the training process. **During training the velocity field and growth function, we don't have to compute any ot plan**, one just need to sample batch $(x_0,x_1)$ from the precomputed ot plan(s) $\pi$ (to train $v$) and sums up the corresponding rows of $x_0$ (to train $g$) **(See Eq.(12), Page5)**. Since the pre-computed plan(s) is on full sample or Big Batches and is fixed during training, we have better training stability compared with other mini-batch ot approaches.
>
> In the following Tables r1-1,r1-2,r1-3,r1-4, we report the sensitivity analysis on $n$ (Table r1-1: number of big batches in precomputing the OT over dataset), batchsize (Table r1-2: batchsize in training loss $\mathcal{L}\_{\mathrm{VGFM}}$ ), sample size (Table r1-3: sampling size in distribution fitting loss $\mathcal{L}\_{\mathrm{OT}}$ ), $n$ (Table r1-4: number of batches in computing distribution fitting loss $\mathcal{L}\_{\mathrm{OT}}$ ), on EB 50D dataset under the same experiment setting of ablation study (Table 3, Page 9).
>
> Table r1-1: Sensitivity analysis of $n$ on EB dataset with metrics (computation time and $W_1$/RME at each time).
>
> |$n$|Time for precomputing OT plan(s)|t1|t2|t3|t4|Average|
> |-|-|-|-|-|-|-|
> |1|10.67|7.951/0.039|8.747/0.042|9.244/0.019|9.620/0.044|8.890/0.036|
> |2|2.96|7.985/0.035|8.823/0.057|9.284/0.035|9.649/0.071|8.935/0.049|
> |5|2.54|7.997/0.038|8.969/0.031|9.436/0.009|9.799/0.042|9.050/0.030|
> |8|2.13|7.970/0.026|8.855/0.070|9.237/0.046|9.536/0.081| 8.899/0.055|
>
> Table r1-2: Sensitivity analysis of batch size in $\mathcal{L}\_{\mathrm{VGFM}}$  on EB dataset with metrics ($W_1$/RME at each time t).
>
> |Batchsize in matching loss|t1|t2|t3|t4|Average|
> |-|-|-|-|-|-|
> |64|7.949/0.034|8.863/0.035|9.250/0.012|9.712/0.047|8.943/0.032|
> |128|7.995/0.023|8.942/0.060|9.227/0.036|9.685/0.069|8.962/0.047|
> |256|7.951/0.039|8.747/0.042|9.244/0.019|9.620/0.044|8.890/0.036|
> |512|7.997/0.037|8.743/0.034|9.150/0.013|9.685/0.045|8.893/0.032|
> |1024|7.989/0.035|8.819/0.059|9.296/0.023|9.738/0.058|8.960/0.043|
> |2048|7.904/0.029|8.800/0.069|9.277/0.041|9.580/0.076|8.890/0.053|
>
> Table r1-3: Comparison of full sampling and mini-batch sampling in $\mathcal{L}\_{\mathrm{OT}}$ on EB dataset with metrics ($W_1$/RME at each time t).
>
> |Sample size|t1|t2|t3|t4|Average|
> |--|--|--|--|--|--|
> |64|8.251/0.032|9.221/0.046|9.745/0.028|10.366/0.057|9.395/0.040|
> |128|8.079/0.030|9.095/0.053|9.625/0.029|9.930/0.057|9.182/0.042|
> |256|8.117/0.032|9.125/0.052|9.758/0.029|9.892/0.057|9.223/0.042|
> |512|8.021/0.034|8.886/0.046|9.184/0.022|9.632/0.048|8.930/0.037|
> |1024|7.978/0.034|8.772/0.044|9.214/0.017|9.519/0.042|8.870/0.034|
> |2048|7.939/0.037|8.807/0.039|9.193/0.016|9.499/0.041|8.859/0.033|
> |full samplling|7.951/0.039|8.747/0.042|9.244/0.019|9.620/0.044|8.890/0.036|
>
>
> Table r1-4: Results of using Big Batch strategy in $\mathcal{L}\_{\mathrm{OT}}$ to speed up computation on EB dataset with metrics ($W_1$/RME at each time t and the training time).
>
> |$n$|t1|t2|t3|t4|Average|Training time(min)|
> |-|-|-|-|-|-|-|
> |1|7.951/0.039|8.747/0.042|9.244/0.019|9.620/0.044|8.890/0.036|13|
> |2|7.938/0.035|8.806/0.040|9.134/0.014|9.545/0.043|8.855/0.033|11|
> |5|7.948/0.034|8.764/0.041|9.153/0.016|9.435/0.045|8.825/0.034|10|
> |8|7.968/0.033|8.774/0.043|9.172/0.017|9.446/0.046|8.840/0.034|10|
>
> These results demonstrate the robustness of performance to the number of big batches in the offline sampling strategy (Table r1-1) and distribution fitting loss $\mathcal{L}\_{\mathrm{OT}}$ (Table r1-4), and the effect of batch size in $\mathcal{L}\_{\mathrm{VGFM}}$ and sampling size in computing distribution-fitting loss $\mathcal{L}\_{\mathrm{OT}}$ (Tables r1-2/3) on the final performance.
>
> ### **Q3: Clarification on if relying on the Sinkhorn algorithm and exploration of Sinkhorn divergence.**
>
> We compute the distribution fitting loss $W_1$ in Section 3.4 using EMD distance by `pot` library, following MIOFlow [Guillaume et al., NeurIPS22] and DeepRUOT [Zhang et al., ICLR25]. Particularly, we compute the OT using the `pot` libary, using the function `pot.emd()` and **only backpropagate the gradient with respect to the cost $c$** in $W_1$ since $\pi$ is solved using a network flow algorithm, which is not compatible with PyTorch’s automatic differentiation. We will make a clear statement in Appendix.
>
>
> As suggested, we replace the EMD distance with Sinkhorn divergence using CUDA/C++ based `geomloss` library. Table r1-5 verified the effectiveness of Sinkhorn divergence for the distribution fitting loss.
>
> Table r1-5: comparison of EMD and sinkhorn divergence (with different entropic regularization parameters) on EB dataset with metrics ($W_1$/RME at each time step).
> |Distribution fitting loss|t1|t2|t3|t4|Average|Training time(min)|
> |-|-|-|-|-|-|-|
> |EMD|7.951/0.039|8.747/0.042|9.244/0.019|9.620/0.044|8.890/0.036|13|
> |$S_{\varepsilon}(\varepsilon=0.001)$|7.902/0.018|8.767/0.013|9.063/0.083|9.507/0.096|8.809/0.052|9|
> |$S_{\varepsilon}(\varepsilon=0.002)$|7.904/0.032|8.791/0.033|9.111/0.104|9.523/0.120|8.832/0.072|9|
> |$S_{\varepsilon}(\varepsilon=0.005)$|7.917/0.048|8.768/0.062|9.086/0.137|9.518/0.151|8.817/0.099|9|
> |$S_{\varepsilon}(\varepsilon=0.01)$|7.904/0.035|8.801/0.038|9.102/0.113|9.511/0.127|8.829/0.078|9|
> |$S_{\varepsilon}(\varepsilon=0.05)$|7.908/0.036|8.787/0.045|9.080/0.120|9.512/0.129|8.821/0.082|9|
> |$S_{\varepsilon}(\varepsilon=0.1)$|7.916/0.030|8.773/0.022|9.111/0.091|9.591/0.108|8.847/0.062|9|
>
> It can be found that the sinkhorn divergence achieves further improvement on the accuracy of dynamics reconstruction, and costs less time for computing the distribution fitting loss. We will include this result of sinkhorn divergence in the appendix and cite it in the discussion section of the manuscript.
>
> ### **Q4: Motivation for time-independent growth function.**
>
> The selection of time-independent growth function is because of
> + **First, it minimizes $L^2$ energy potential.** As we have discussed in Appendix A.4, time-independent growth function is not only easy to implement but also minimizes the $L^2$ energy potential, which satisfies the principle of least action.
> + **Second, it can be explained by the Malthusian growth model**, i.e., the exponential growth model. The Malthusian growth model is $\frac{\mathrm{d}p_t}{\mathrm{d}t}=gp_t$, where $p_t$ is the population size and $g$ is a constant growth rate. Our model on the growth $\frac{\partial p_t}{\partial t}=g_tp_t$ (refer to Eq.5, Page 4, Line 145-146) is consistent with the above Malthusian growth model. **In these models, g is treated as a constant** based on the assumption that the resources are abundant and the environment is stable, causing growth rates to remain relatively stable over time. In the context of scRNA-seq experiments, these conditions are **typically satisfied** because cells are often cultured or sampled under controlled laboratory conditions, where nutrient supply, temperature, and other environmental factors are maintained at constant levels. Thus, choosing this form is biologically reasonable, especially when there is no prior knowledge, such as the maximum carrying capacity of the environment.
>
> ### **Q5: On notations.**
>
> Thanks for your suggestion. Upon revisiting our paper, we realized that the discrete entropic semi-relaxed optimal transport formulation in Eq. (11) can be removed, as it is a standard result. Accordingly, we can directly use $\pi^*$ in place of $\pi^{0\to1}$. In addition, we can describe the barycentric mapping and $N_i$ in plain language (line 188) for better clarity. We will also carefully review the manuscript to refine the notations and formulations throughout.

---

> ### Author Response · Authors · 2025-08-05
> **Kindly reminding the discussion deadline**
>
> Dear Reviewer kHRn,
>
> We would like to thank you again for the valuable comments and suggestions. We have responded to the comments in detail and hope to address your concerns. As the discussion period is drawing to a close, we kindly remind you that if you have other concerns, please let us know. We will try our best to clarify further.
>
> Best regards,
>
> The Authors

---

### Decision · Program_Chairs · 2025-09-17

**Decision:**

Accept (poster)

**Comment:**

This paper presents velocity-growth flow matching (VGFM) a method for modeling single-cell population dynamics that handles mass imbalance using semi-relaxed optimal transport. This is a crucial challenge in the single-cell modeling problem. The consensus among reviewers is to accept this paper which I agree with.

While one reviewer (kHRn) expressed concerns during the initial review, that reviewer did not engage in the discussion despite a large number of reminders and will therefore be discounted. Otherwise, the rebuttal discussion went smoothly and improved the opinion of two reviewers. I recommend acceptance at this time.